# Decoding the m⁶A epitranscriptomic landscape for biotechnological applications using a direct RNA sequencing approach

Chuwei Liu [1,7], Heng Liang[2,7], Arabella H. Wan[3,7], Min Xiao[2,7], Lei Sun[2,7], Yiling Yu[4], Shijia Yan [2], Yuan Deng[2], Ruonian Liu[2], Juan Fang[5], Zhi Wang[5] ✉, Weiling He [1,6] ✉ & Guohui Wan [2] ✉

Epitranscriptomic modifications, particularly N6-methyladenosine (m⁶A), are crucial regulators of gene expression, influencing processes such as RNA stability, splicing, and translation. Traditional computational methods for detecting m⁶A from Nanopore direct RNA sequencing (DRS) data are constrained by their reliance on experimentally validated labels, often resulting in the underestimation of modification sites. Here, we introduce pum6a, an innovative attention-based framework that integrates positive and unlabeled multi-instance learning (MIL) to address the challenges of incomplete labeling and missing read-level annotations. By combining electrical signal features with base alignment data and employing a weighted Noisy-OR probability mechanism, pum6a achieves enhanced sensitivity and accuracy in m⁶A detection, particularly in low-coverage loci. Pum6a outperforms existing methods in identifying m⁶A sites across various cell lines and species, without requiring extensive parameter tuning. We further apply pum6a to study the dynamic regulation of m⁶A demethylases in gastric cancer under hypoxia, revealing distinct roles for FTO and ALKBH5 in modulating m⁶A modifications and uncovering key insights into m⁶A -mediated transcript stability. Our findings highlight the potential of pum6a as a powerful tool for advancing the understanding of epitranscriptomic regulation in health and disease, paving the way for biotechnological and therapeutic applications.

The discovery and subsequent exploration of RNA nucleotide modifications[1,2], particularly $N^6$-methyladenosine (m⁶A), have unveiled a complex layer of gene expression regulation with profound implications for biotechnology and therapeutic development[3–6]. As the most prevalent internal modification in mammalian mRNA[7,8], m⁶A is intricately regulated through the dynamic interplay of its deposition by a methyltransferase complex consisting of METTL3, METTL14, WTAP, KIAA1429, and RBM15[9–14], and its removal by demethylases

[1]Department of Gastrointestinal Surgery, The First Affiliated Hospital, Sun Yat-Sen University, Guangzhou 510080, China. [2]National-Local Joint Engineering Laboratory of Druggability and New Drug Evaluation, National Engineering Research Center for New Drug and Druggability (cultivation), Guangdong Province Key Laboratory of New Drug Design and Evaluation, School of Pharmaceutical Sciences, Sun Yat-Sen University, Guangzhou 510006, China. [3]Department of Medicine, Keck School of Medicine, University of Southern California, Los Angeles, CA 90033, USA. [4]School of Public Health, Sun Yat-Sen University, Guangzhou 510080, China. [5]Hospital of Stomatology, Guanghua School of Stomatology, Guangdong Provincial Key Laboratory of Stomatology, Sun Yat-Sen University, Guangzhou 510055, China. [6]Department of Gastrointestinal Surgery, Xiang'an Hospital of Xiamen University, School of Medicine, Xiamen University, XIamen 361000, China. [7]These authors contributed equally: Chuwei Liu, Heng Liang, Arabella H. Wan, Min Xiao, Lei Sun. ✉e-mail: wangzh75@mail.sysu.edu.cn; hewling@mail.sysu.edu.cn; wanguoh@mail.sysu.edu.cn

FTO[15] and ALKBH5[16]. This modification exerts a profound influence on various aspects of RNA biology, including its structure[17], stability[18], splicing[19], and translation[20], underscoring its potential as a target for biotechnological innovations.

Over the past decade, technological advancements have significantly improved our ability to map m⁶A sites transcriptome-wide. Initially, antibody-based techniques like MeRIP-Seq[21], m⁶A-Seq[22] and miCLIP[23] offer the first glimpses into the m⁶A epitranscriptome but are limited by low resolution and high false-positive rates due to non-specific antibody interactions[24]. These methods often require complementary approaches, such as chemical[7,25,26] and enzymatic[27–29] assays, to increase specificity, but they still struggle with the accurate detection of m⁶A at single-nucleotide resolution. In response to these challenges, newer methodologies have been developed. For instance, miCLIP2[30], combined with the m6Aboost machine learning algorithm, represents a significant improvement in m⁶A site detection. miCLIP2 refines antibody specificity and increases the library complexity, allowing for better mapping of m⁶A sites, even those outside the canonical DRACH motif. This innovation is particularly valuable as it uncovers previously unrecognized m⁶A sites that could play crucial roles in RNA metabolism and disease.

Parallel to these experimental advances, computational tools have evolved to address the limitations of traditional m⁶A detection methods. Direct RNA sequencing (DRS) technologies, such as those developed by Oxford Nanopore Technologies (ONT)[31], have revolutionized the field by allowing the sequencing of native RNA molecules without prior conversion to cDNA, thus preserving RNA modifications like m⁶A. However, the interpretation of DRS data is complex, requiring sophisticated computational approaches to distinguish m⁶A-induced signal variations from other noise in the sequencing data[7]. The introduction of deep learning and multi-instance learning (MIL) frameworks, such as m6Anet[32], marks a substantial leap forward. m6Anet leverages MIL to account for the heterogeneity in RNA samples, where not all reads at a given site are modified. By learning high-dimensional representations of individual reads before aggregating them, m6Anet significantly improves the prediction accuracy of m⁶A sites across diverse biological contexts. Nonetheless, while m6Anet has shown robustness in detecting m⁶A, it still faces challenges such as dependency on high-quality training labels and the potential underestimation of m⁶A sites due to variable read coverage[32].

Despite these advancements, several limitations persist. Traditional m⁶A detection methods are often constrained by the need for negative controls, reliance on specific sequence motifs, and the inability to accurately quantify m⁶A at low stoichiometry sites. Moreover, current computational approaches may not fully capture the complexity of m⁶A modifications in heterogeneous RNA populations, leading to missed detections in low-abundance transcripts[30,32]. Additionally, most existing models are designed for specific sequence contexts, limiting their generalizability across different species or cell types. In response to the limitations of existing m⁶A detection methods, we develop pum6a, an innovative computational framework that employs an attention-based positive and unlabeled multi-instance learning strategy to enhance the detection of m⁶A sites from direct RNA-Seq data. This model is specifically designed to accurately identify m⁶A modifications, even without comprehensive, experimentally validated labels. It achieves high sensitivity and specificity, especially at sites with low modification frequencies, and is adaptable to a broad range of RNA sequences and biological conditions. Differing from m6Anet, which aggregates features from reads at a specific site, pum6a introduces an attention mechanism that selectively focuses on the most informative reads, significantly improving the signal-to-noise ratio. This refinement not only increases the accuracy of m⁶A site identification but also broadens the model's applicability to detect other RNA modifications, making it a versatile tool for epitranscriptomic research.

## Results

### Enhancement of anomaly detection through the pum6a framework

The pum6a framework introduces a significant advancement in anomaly detection by addressing the challenges inherent in positive and unlabeled multi-instance learning. As shown in Fig. 1a (Methods), pum6a utilizes a structured, multi-module approach, beginning with an instance-level feature extraction phase where each data instance is transformed into compact, low-dimensional embeddings. This is followed by a self-attention mechanism that offers a flexible alternative to conventional aggregation methods, such as max or mean pooling. The self-attention module incorporates a trainable weighted average, implemented via a dual-layer neural network, enabling the adaptive and interpretable aggregation of features across instances[33].

Next, these aggregated instance features are processed by a classifier module to predict the probability of a positive class within each bag. The self-attention mechanism enhances interpretability by identifying key instances that contribute most significantly to the classification outcome. To further refine multi-instance learning, Platt scaling is applied to the attention scores, converting them into instance-level probabilities. These probabilities are then integrated into a bag-level classification using a weighted Noisy-OR function[34], which effectively combines the contributions of individual instances.

To mitigate potential overfitting toward positive instances, we implemented a balancing strategy by ensuring the number of selected negative bags equals the number of positive bags ($|R| = |P|$). This balanced sampling approach reduces bias toward positive classifications and enhances the model's robustness and generalization across various datasets. As a result, pum6a demonstrates improved classification performance, offering a more accurate and balanced approach to anomaly detection in MIL settings.

### The pum6a framework outperforms the state-of-the-art competitors

The pum6a framework demonstrated superior performance in multi-instance anomaly detection when compared to several state-of-the-art methods, including PUMA[34] and the Inexact Autoencoder (IAE)[35], as well as other baselines like Random Forest and puIF (an unsupervised Isolation Forest with logistic regression). We evaluated pum6a across a range of datasets, including a modified MNIST image dataset and 20 well-established benchmarks for anomaly detection. To ensure robustness and generalizability, the model was tested using a stratified 5-fold cross-validation procedure, with varying label frequencies (10% to 50%), and a weighted Noisy-OR method for consistent interpretation (Fig. 1a, Methods).

The results highlight pum6a's ability to accurately identify and prioritize critical instances, as demonstrated in the MNIST dataset, where images of the digit "9" consistently received the highest attention, suggesting the framework's proficiency in capturing salient features that are critical for effective anomaly detection (Fig. 1b). When benchmarked against other methods, pum6a exhibited superior adaptability and accuracy, particularly under varying label frequencies. The framework achieved consistently higher ROC AUC scores at both the bag and instance levels across all conditions (Bag level ROC AUC: 10%LF: $0.60 \pm 0.15$, 20%LF: $0.73 \pm 0.10$, 30%LF: $0.77 \pm 0.10$, 40%LF: $0.81 \pm 0.08$, 50%LF: $0.88 \pm 0.06$; Instance level ROC AUC: 10%LF: $0.66 \pm 0.14$, 20%LF: $0.76 \pm 0.11$, 30%LF: $0.81 \pm 0.09$, 40%LF: $0.86 \pm 0.06$, 50%LF: $0.90 \pm 0.04$) (Fig. 1c, Supplementary Data 1).

To further evaluate pum6a's generalizability, we applied the framework to 20 benchmark datasets spanning various domains of anomaly detection. The results consistently confirmed pum6a's superior performance, with average ROC AUC scores exceeding 0.70 across different label frequency settings (Fig. 2a, b, Supplementary Data 1). Notably, pum6a maintained robust performance even as dataset complexity increased, particularly in high-dimensional

 

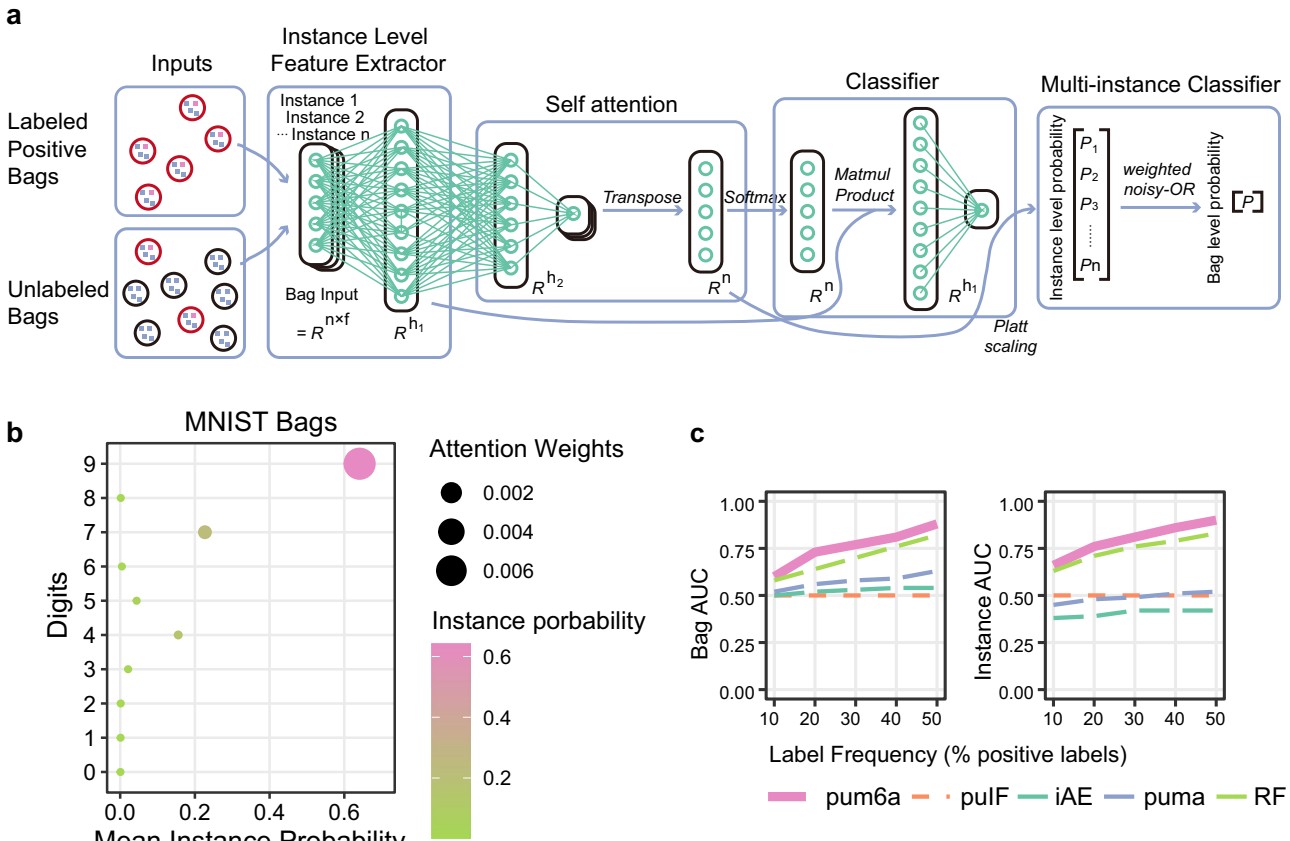

**Fig. 1 | Overview of the pum6a framework and its evaluation on the MNIST bag dataset. a** Schematic of the pum6a framework, highlighting its key components for multi-instance learning and m⁶A site detection. **b** Dot plot illustrating attention weights and instance probabilities in positive bags. Dot size corresponds to attention weight, and color intensity represents the probability of specific digit identification, with the model focusing on digits 9, 7, and 4. **c** Comparative ROC AUC performance metrics of pum6a versus baseline models at varying label frequencies. Left: Bag-wise ROC AUC; Right: Instance-wise ROC AUC, demonstrating pum6a's superior adaptability across complex datasets.

settings. The aggregate ranking across all experiments revealed that pum6a achieved the lowest (and therefore best) average rank, affirming its competitive edge and versatility in anomaly detection tasks (Fig. 2c, d). These findings establish pum6a as a highly effective tool for multi-instance learning in positive and unlabeled setting, demonstrating its potential to significantly improve anomaly detection performance across a wide range of datasets and applications.

### Refining the pum6a framework for precise detection of m⁶A sites

Tailoring the pum6a framework for the detection of m⁶A modifications within direct RNA sequencing data revealed its capacity to accurately distinguish signal variations and base-calling errors generated by ONT nanopore technology (Fig. 3a, Methods)[8,36]. This process involved a comprehensive preprocessing pipeline, including base calling, re-squiggling, and alignment using minimap, which transformed raw signal data into structured 'site bags' for further analysis (Methods).

We trained and validated pum6a using two datasets from HEK293T cell lines provided by the Singapore Nanopore Expression Project[37]—replicate 1 for training and replicate 4 for validation. pum6a's performance was compared against six established m⁶A detection methods, including EpiNano, MINES, m6Anet, ELIGOS, Nanom6A, and Tombo. To ensure a fair comparison, we applied filtering thresholds of 3, 5, and 20 reads per site, as both ELIGOS and m6Anet discard sites with fewer than 5 and 20 reads, respectively. Experimentally validated m⁶A sites from CIMS, CTIS, m6ACE, GLORI served as additional reference[7,23,38], addressing potential label underestimation (Fig. 3b).

Our benchmark analysis, based on ROC and precision-recall (PR) curves (Fig. 3c–h), demonstrated pum6a's superior performance across all thresholds. The ROC AUC scores increased from 0.826 with >3 reads per site to 0.842 with >20 reads per site, while PR AUC scores improved from 0.580 to 0.615. Across all filtering criteria, pum6a outperformed the other methods, consistently providing higher accuracy in identifying m⁶A sites (Fig. 3i, Supplementary Data 2). Furthermore, pum6a's precision in identifying the top 18,000 m⁶A sites remained stable, outperforming competing methods across various conditions, highlighting its robustness in m⁶A site prediction (Fig. 3j). These findings underscore pum6a's effectiveness in leveraging training and validation datasets to optimize anomaly detection in m⁶A site identification.

To further evaluate pum6a's generalization capabilities, we extended our analysis to mouse embryo stem cells (mESCs), utilizing miCLIP and miCLIP2 data as the ground truth for m⁶A site labeling[30,39]. pum6a again demonstrated superior performance, with ROC AUC scores increasing from 0.810 with >3 reads per site to 0.827 with >20 reads, and PR AUC scores increasing from 0.325 to 0.490 (Fig. 4b–g). Notably, pum6a identified a higher number of modified sites with greater accuracy than baseline methods, further affirming its precision (Fig. 4n, Supplementary Data 3). Interestingly, the minimal overlap in m⁶A site predictions between species suggests species-specific modification patterns, emphasizing pum6a's adaptability and biological relevance. The alignment of five-mer profiles from pum6a predictions with experimental data further supports this finding (Figs. 4i, f, S1). Additionally, pum6a achieved exceptional results on a constructed dataset, with ROC AUC reaching 0.946 and PR AUC 0.952 at the bag-

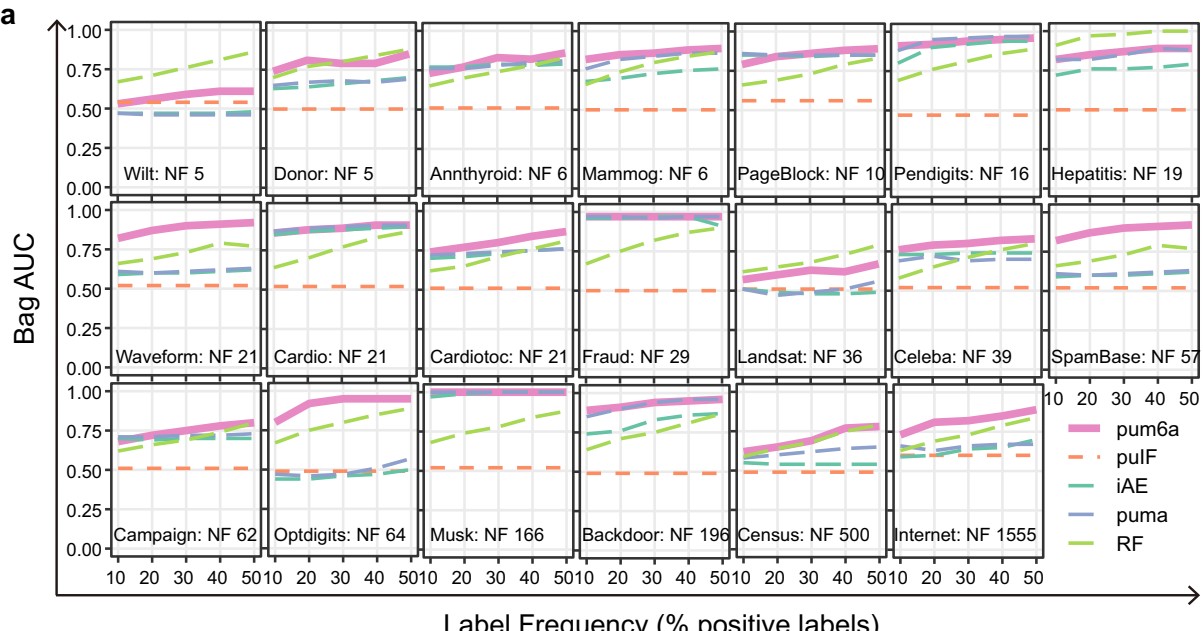

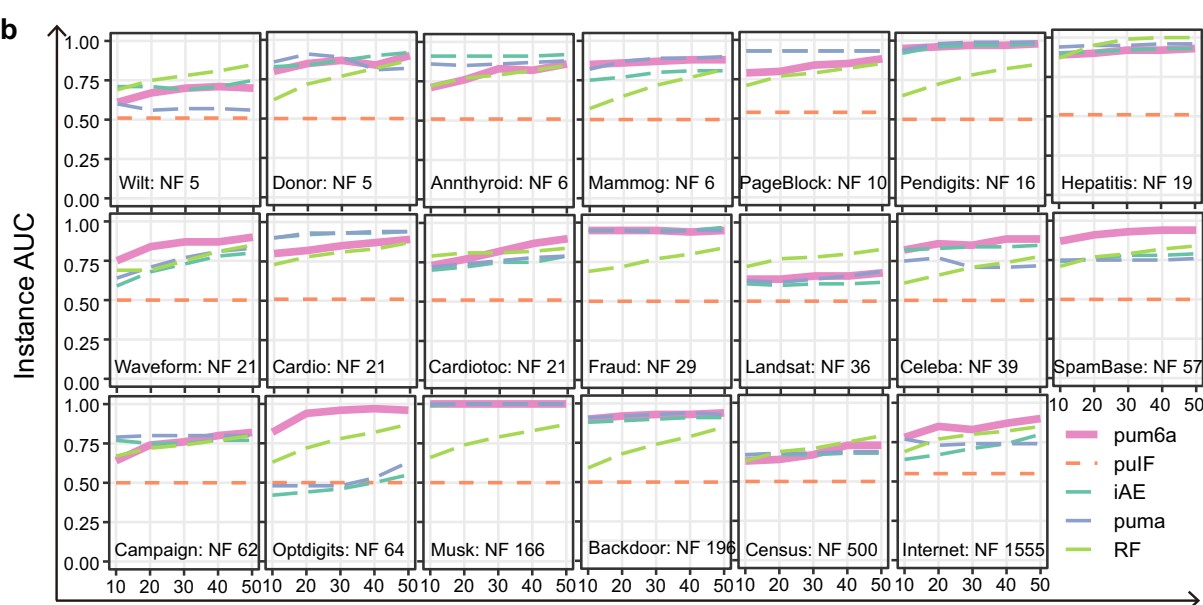

**c**

| | Bag ranks (avg. ± std.) | | | | |
|---|---|---|---|---|---|
| c% | iAE | puIF | pum6a | PUMA | RF |
| 10 | 3.09± 1.05 | 4.52± 0.88 | **2.00± 1.23** | 2.43± 1.15 | 2.97± 1.28 |
| 20 | 3.13± 1.04 | 4.60± 0.82 | **1.83± 1.07** | 2.56± 1.20 | 2.87± 1.22 |
| 30 | 3.18± 1.00 | 4.65± 0.75 | **1.78± 1.04** | 2.61± 1.19 | 2.79± 1.20 |
| 40 | 3.27± 1.01 | 4.69± 0.70 | **1.80± 1.00** | 2.65± 1.18 | 2.60± 1.16 |
| 50 | 3.39± 0.95 | 4.75± 0.63 | **1.80± 0.89** | 2.74± 1.14 | 2.33± 1.16 |

**d**

| | Instance ranks (avg. ± std.) | | | | |
|---|---|---|---|---|---|
| c% | iAE | puIF | pum6a | PUMA | RF |
| 10 | 2.55± 1.06 | 4.81± 0.51 | **2.24± 1.25** | 2.17± 1.05 | 3.23± 1.07 |
| 20 | 2.66± 1.04 | 4.86± 0.44 | **2.15± 1.18** | 2.30± 1.11 | 3.03± 1.11 |
| 30 | 2.68± 1.05 | 4.89± 0.39 | **2.11± 1.12** | 2.39± 1.10 | 2.93± 1.16 |
| 40 | 2.70± 1.05 | 4.90± 0.37 | **2.09± 1.12** | 2.44± 1.09 | 2.87± 1.16 |
| 50 | 2.71± 1.05 | 4.92± 0.30 | **2.06± 1.11** | 2.52± 1.08 | 2.80± 1.17 |

**Fig. 2 | Comparative performance of pum6a and baseline models across diverse datasets. a, b** ROC AUC scores of pum6a and baseline models on 20 different datasets. **a** Bag-wise ROC AUC; **b** Instance-wise ROC AUC, showing pum6a's consistent performance across various biological data. **c, d** Average rank of each method across all experiments and different label frequencies. **c** Bag level; **d** Instance level ranking, with pum6a outperforming other methods in predictive accuracy and model robustness.

level, and ROC AUC 0.857 and PR AUC 0.893 at the read level. These results solidify pum6a's capability for precise m⁶A site detection in ONT direct RNA sequencing data, demonstrating its broad applicability and superior accuracy across various cell lines and species (Fig. 4h–k).

We observed that both pum6a and m6aNet, two advanced deep-learning models, exhibited robust performance in predicting m⁶A modifications within third-generation sequencing tasks. Given that m6aNet's implementation specifically includes a prediction threshold

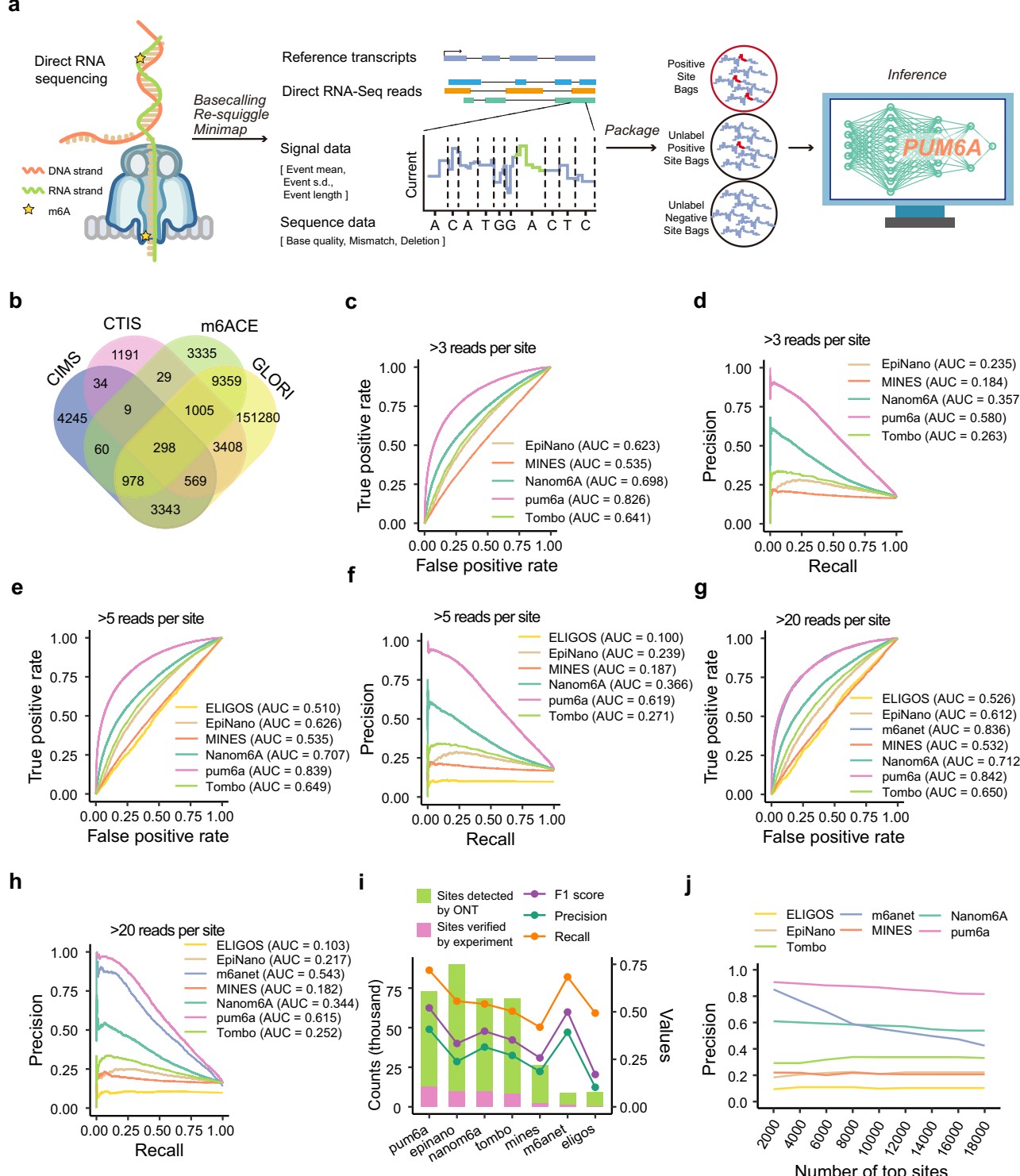

**Fig. 3 | Application of pum6a for m⁶A detection in ONT direct RNA sequencing data in HEK293T cells. a** Schematic diagram of the pum6a model tailored for m⁶A detection from ONT direct RNA sequencing. The conceptual structure of the workflow was inspired by previous works by Hendra et al. (Nature Methods)[32] and Zhong et al. (Nature Communications)[47], and independently designed and integrated. This figure was adapted from Hendra et al. (Nature Methods, 2022, https://doi.org/10.1038/s41592-022-01666-1) and Zhong et al. (Nature Communications, 2023, https://doi.org/10.1038/s41467-023-37596-5), both published under a CC-BY license (https://creativecommons.org/licenses/by/4.0/). Modifications were made.

**b** Distribution of m⁶A modification sites identified in HEK293T cells across four experiment protocols. **c, d** Comparison of pum6a's performance with EpiNano, MINES, Nanom6A, and Tombo using ROC (**c**), and PR curves (**d**) for datasets with at least 3 reads. **e, f** ROC (**e**), and PR curves (**f**) for datasets with at least 5 reads, comparing pum6a with additional methods including ELIGOS. **g, h** ROC (**g**), and PR curves (**h**) for datasets with at least 20 reads, incorporating m6anet into the comparison. **i** Summary of precision, recall, and F1 scores for all evaluated models. **j** Precision analysis of the top 18,000 m⁶A sites across four protocols, showing pum6a's superior precision.

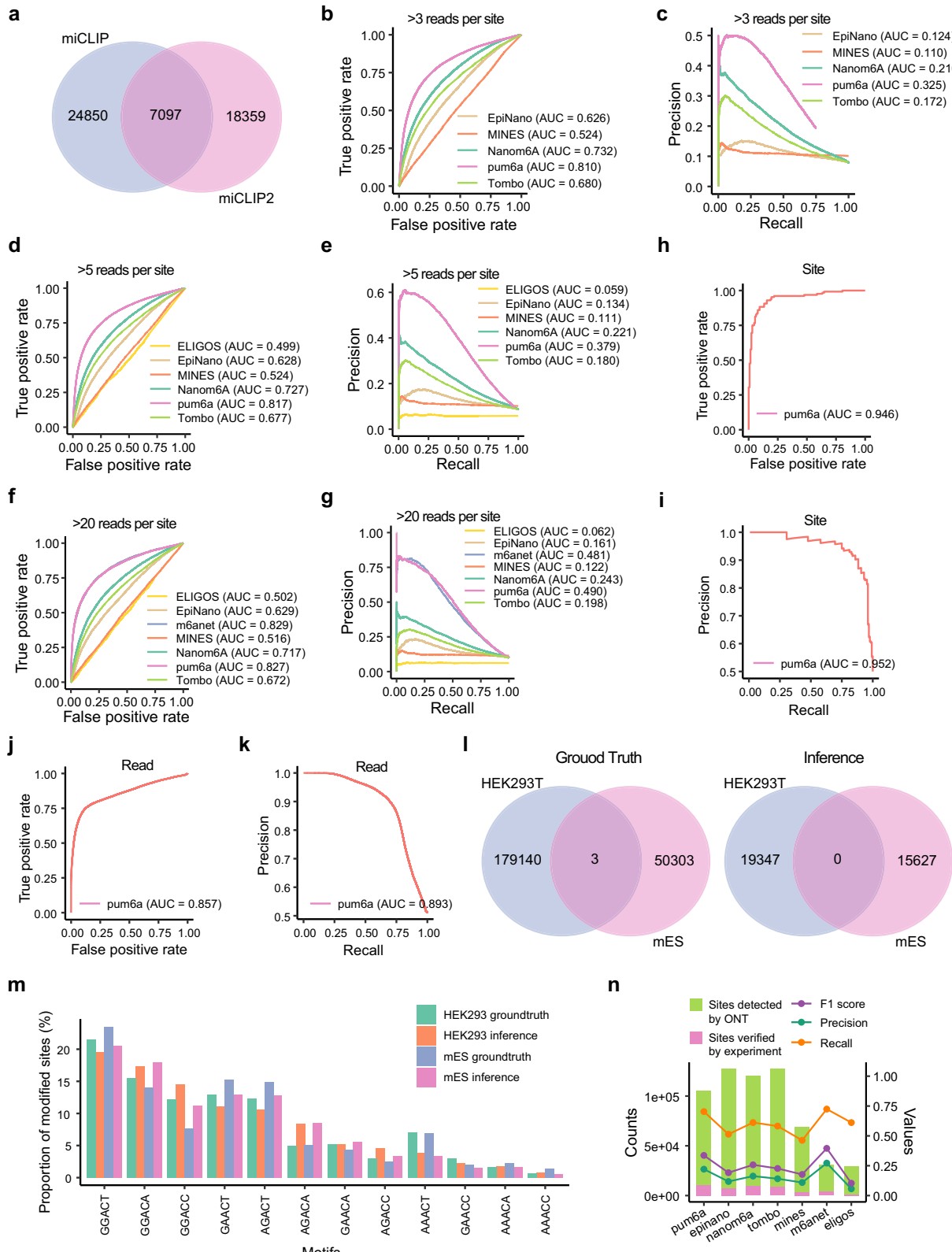

**Fig. 4 | Evaluation of pum6a on mouse embryonic stem cells and synthetic datasets. a** Distribution of m⁶A modification sites identified in HEK293T cells across two experiment protocols. **b, c** ROC (**b**), and PR curves (**c**) comparing pum6a to baseline models on datasets with at least 3 reads. **d, e** ROC (**d**), and PR curves (**e**) for datasets with at least 5 reads. **f, g** ROC (**f**), and PR curves (**g**) for datasets with at least 20 reads, showcasing pum6a's performance relative to other models. **h, i, j, k** ROC (**h, j**), and PR (**i, k**) curves on synthetic data at the bag and instance levels, demonstrating pum6a's accuracy. **l** Comparison of m⁶A modification sites distributions between species (HEK293T and mouse embryonic stem cells). Left: Ground truth set obtained from experiment protocol; Right: Pum6a inference set. **m** Proportion of m⁶A modification sites predicted by pum6a and experimental protocols for RRACH motifs in both species. **n** Summary of precision, recall, and F1 scores across two protocols, highlighting pum6a's strong performance.

based on a 20 reads per site criterion, we were motivated to undertake a comparative analysis of both models across varying read thresholds to further clarify performance distinctions, particularly between pum6a and m6aNet. To this end, we adapted the original m6aNet code to conduct a detailed evaluation under 0, 3, and 5 reads per site thresholds. The comparative results showed that both models maintained strong predictive capabilities across all tested thresholds. Moreover, both pum6a and m6aNet showed improved prediction accuracy with higher read thresholds. Notably, pum6a consistently outperformed m6aNet at each threshold level, particularly achieving superior results in terms of area under the receiver operating characteristic curve (ROC AUC) and area under the precision-recall curve (PR AUC) at the 0 threshold condition (Fig. S1a-f).

This observation prompted us to investigate the essential components contributing to the pum6a model's performance. To elucidate these contributions, we performed targeted modifications by replacing the attention layer and the weighted-Noisy-OR layer in pum6a independently, then evaluated the impacts on performance. Experimental results indicated that removing the attention layer had minimal effect on model accuracy, whereas replacing the weighted Noisy-OR layer with a standard Noisy-OR layer led to a notable decrease in performance (Fig. S1g-h). These findings suggest that, within third-generation sequencing applications, a fully connected layer alone is highly effective; combining it with either a Noisy-OR layer (as in m6aNet) or a weighted-Noisy-OR layer (as in pum6a) yields high predictive accuracy for base modifications. Notably, our model framework demonstrates a significant drop in accuracy when replacing pum6a's weighted Noisy-OR with alternative components, indicating its critical role in performance.

The pum6a model represents an enhanced iteration of the puma model[34], originally challenged by complex datasets. To address this limitation, we integrated an attention layer, inspired by feature aggregation methods suggested by Ilse et al.[33,34], which provided a mechanism for better feature aggregation in pum6a. This modification led to improved performance on complex datasets, while maintaining comparable results on simpler datasets (Fig. 1b, c). Building on previous work, our approach also incorporated essential electrical signal features and alignment data specific to individual modification sites, totaling 40 features in all. While our third-generation sequencing dataset is relatively manageable in complexity, we anticipate that pum6a holds significant promise for applications involving more complex biological data.

## Dynamic m6A modification in gastric cancer mediated by m6A demethylases under hypoxia stress

The dynamic regulation of m6A modifications by demethylases FTO and ALKBH5 under hypoxia plays a critical role in the adaptation of gastric cancer cells to oxygen-limited environments[40,41]. To explore this relationship, we employed knockdown strategies in gastric cancer cell lines under normoxic (20% $O_2$) and hypoxic (1% $O_2$) conditions, followed by ONT direct RNA sequencing to analyze m6A modifications and gene expression. Poly(A) RNA was isolated for high-resolution analysis (Fig. 5a), and knockdown efficiency was validated by Western blot and direct RNA sequencing (Fig. 5b, c). The results revealed a distinct regulatory pattern under hypoxia: ALKBH5 expression was significantly upregulated, whereas FTO expression remained unchanged (Fig. 5c, d, Supplementary Data 4), suggesting a pivotal role for ALKHB5 in hypoxic response while FTO may operate independently of oxygen levels.

Hypoxia reduced proliferation in AGS cells (Fig. 5e), while MKN28 cells exhibited greater tolerance, showing a milder reduction in growth (Fig. S2b), consistent with previous studies indicating tumor adaptation to hypoxic environments[42]. Further investigation into the roles of FTO and ALKBH5 under varying oxygen conditions revealed that overexpression of FTO enhanced proliferation in AGS cells under

normoxia but had no significant effect under hypoxia or in MKN28 cells (Figs. 5f, g, S2c, d). Overexpression of ALKBH5 did not impact cell growth in either cell line. However, knockdown of either FTO or ALKBH5 led to a significant reduction in proliferation in both cell lines, particularly under hypoxia, as confirmed by cell counts and Edu assays (Fig. 5h–k, l, m; Fig. S3e–h, i). These findings underscore the essential roles of FTO and ALKBH5 in maintaining gastric cancer cell growth under oxygen-limited conditions.

To further elucidate the molecular mechanisms involved, we employed the pum6a framework to perform site-specific analysis of m6A modifications, comparing knockdown samples to controls under both normoxic and hypoxic consitions. This "Double check" approach identified 1061 and 1686 m6A sites regulated by both demethylases under normoxia and hypoxia, respectively, indicating complementary roles for FTO and ALKBH5 in modulating m6A methylation (Fig. 5n–o, Supplementary Data 5). Analysis of these sites revealed significant shifts in metabolic pathways and cell cycle regulation under both oxygen conditions, with 650 sites altered under normoxia and 1275 under hypoxia, suggesting a role for m6A in metabolic adaptation during hypoxic stress (Fig. 5p, Supplementary Data 6). KEGG pathway enrichment analysis of normoxia-regulated sites highlighted metabolic pathways, while GO analysis emphasized the importance of these sites in cell cycle and division processes, aligning with observed effects on cell proliferation (Fig. 5q, r, h-m; Supplementary Data 7).

Moreover, knockdown of FTO or ALKBH5 led to significant reductions in ATP levels in both AGS and MKN28 cells, under both normoxic and hypoxic conditions (Fig. 5s, t; S2j, k). Similarly, $NAD^+$ levels and the $NAD^+$/NADH ratio were significantly decreased under hypoxia following demethylases knockdown (Fig. 5u, v; S2l, m), suggesting that FTO and ALKBH5 are key regulators of energy homeostasis in gastric cancer cells. These findings indicate that the depletion of FTO and ALKBH5 disrupt cellular metabolic balance, particularly under hypoxic conditions, and highlight their roles in maintaining energy production and supporting cancer cell survival during oxygen deprivation.

## Hypoxia-induced regulation of CXCL10 by m6A demethylases in gastric cancer cells

The tumor-immune microenvironment in gastric cancer, influenced by m6A modifications, is intricately modulated under hypoxic conditions, with CXCL10 emerging as key mediator in tumor-immune interactions. Depletion of m6A demethylases, particularly ALKBH5, under hypoxia leads to the upregulation of CXCL10, which may impact tumor progression and immune response dynamics[43,44]. Using the pum6a framework, we identified several predicted m6A modification sites within the CXCL10 mRNA in AGS cells following ALKBH5 knockdown (Fig. 6a). This knockdown significantly increased CXCL10 mRNA levels in both AGS (Fig. 6b, c) and MKN28 cells (Fig. 6d, e) under hypoxic conditions. Consistent with mRNA results, ELISA assays confirmed that ALKBH5 depletion elevated CXCL10 protein levels, particularly under hypoxia (Fig. 6j–m), indicating that ALKBH5 serves as a negative regulator of CXCL10 expression.

In contrast, FTO knockdown displayed cell-line-specific effects on CXCL10 expression. In AGS cells, FTO knockdown increased CXCL10 mRNA and protein levels under hypoxia (Fig. 6f, g, k), while in MKN28 cells, it led to a downregulation of CXCL10 under the same conditions (Fig. 6h, i, m). These results suggest divergent roles for FTO in regulating CXCL10, promoting its expression in AGS cells and suppressing it in MKN28 cells, thereby highlighting the context-dependent function of FTO in different gastric cancer cell lines.

Further mechanistic insights were obtained through Actinomycin D chase assays, which demonstrated that ALKBH5 knockdown significantly stabilized CXCL10 mRNA in both AGS and MKN28 cells under hypoxia (Fig. 6n, o, p, q). FTO knockdown similarly reduced the decay rate of CXCL10 mRNA in AGS cells under hypoxia, enhancing mRNA

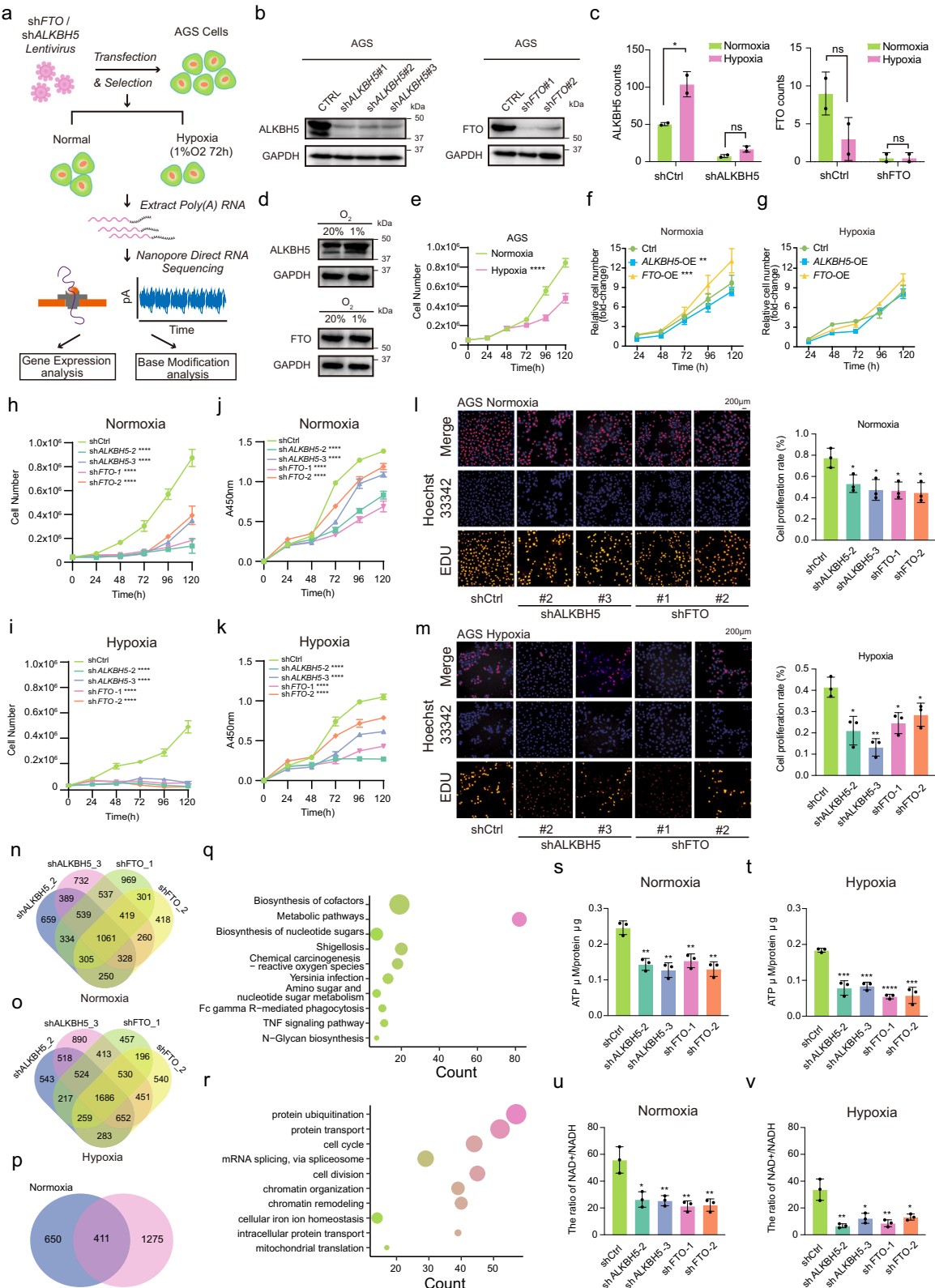

stability (Fig. 6r, s). In MKN28 cells, however, FTO knockdown increased CXCL10 mRNA stability under normoxia but had a diminished effect under hypoxia (Fig. 6t, u). These findings indicate that both ALKBH5 and FTO regulate CXCL10 mRNA stability in a cell-line and oxygen-dependent manner.

To further explore the differential m6A regulation between AGS and MKN28 cells, we analyzed the copy numbers of key m6A writers,

erasers, and readers, revealing that MKN28 cells harbor a heterozygous deletion of the m6A reader YTHDF1 (Fig. S4a, b). YTHDF1 has been shown to promote mRNA translation and stability[45], while YTHDF2 primarily facilitates mRNA decay[46]. In AGS cells, where both readers are intact, YTHDF1 plays a prominent role in stabilizing CXCL10 mRNA, as evidenced by the increased stability following ALKBH5 knockdown (Fig. 6n, o). In contrast, MKN28 cells, with

**Fig. 5 | Dynamic regulation of m⁶A modification by hypoxia and m⁶A deme-thylases in gastric cancer cells. a** Experimental workflow for assessing dynamic m⁶A modification under hypoxia and the effects of m⁶A demethylases. **b**, Validation of ALKBH5 and FTO knockdown efficiency by Western blot in AGS gastric cancer cells. **c** Gene count of ALKBH5 and FTO in AGS cells under different oxygen conditions. *$p$ = 0.0475 by a t-test. **d** Western blot analysis showing differential expression of ALKBH5 and FTO in AGS cells under normoxic and hypoxic conditions. **e** AGS cell proliferation responses to hypoxia, showing suppressed growth. ****$p$ < 0.0001 by a two-way ANOVA. **f, g** Impact of ALKBH5 or FTO overexpression on AGS cell growth under normoxia (**f**) and hypoxia (**g**), quantified by cell count. In **f**, ALKBH5-OE (**$p$ = 0.0019), FTO-OE (***$p$ = 0.0006) by a two-way ANOVA. **h, i, j, k** Effects of ALKBH5 and FTO knockdown on AGS cell proliferation and growth under normoxia (**h, j**) and hypoxia (**i, k**), quantified through cell count (**h–i**) and growth rate (**j–k**) measurements. ****$p$ < 0.0001 by a two-way ANOVA. **l,m** Knockdown of ALKBH5 or FTO significantly reduced proliferation of AGS cells under hypoxia, as measured by EdU assay. Scale bars are 200 μm. Quantification of fold changes was performed using ImageJ. **l**: Normoxia; *$p$ = 0.0268, *$p$ = 0.0174, *$p$ = 0.0118, *$p$ = 0.0124, by a t-test. **m**: Hypoxia; *$p$ = 0.0126, **$p$ = 0.0014, *$p$ = 0.0128, *$p$ = 0.0359, by a t-test. **n, o** Distribution analysis of m⁶A-modified sites in AGS cells following ALKBH5 or FTO knockdown under normoxia (**n**) and hypoxia (**o**).

**p** Overlap analysis showing common m⁶A-modified sites regulated by ALKBH5 and FTO under varying oxygen levels. **q** KEGG pathway enrichment analysis of m⁶A-modified sites regulated by m⁶A demethylases under normoxic conditions. Dot color indicates the number of genes, and dot size represents the −log10 $p$-value of the pathway term. **r** Gene Oncology (GO) enrichment analysis of m⁶A-modified sites regulated by m⁶A demethylases under hypoxia. Dot color indicates the number of genes, and dot size represents the −log10 $p$-value of the biological process term. Statistical analysis (**q, r**) was performed using a two-sided hyper-geometric test with adjustment for multiple comparisons using the Benjamini-Hochberg (BH) method to control the false discovery rate (FDR). **s, t** Significant reduction of ATP levels in AGS cells following FTO/ALKBH5-knockdown under normoxic (**s**) and hypoxic (**t**) conditions. In **s**, **$p$ = 0.0022, **$p$ = 0.0018, **$p$ = 0.0041, **$p$ = 0.002, by a t-test. In **t**, ***$p$ = 0.001, ***$p$ = 0.0001, ****$p$ < 0.0001, ***$p$ = 0.0007, by a t-test. **u, v** Decreased NAD+ levels and NAD + /NADH ratio in AGS cells after FTO/ALKBH5 depletion under normoxic (**u**) and hypoxic (**v**) conditions. In **u**, *$p$ = 0.011, **$p$ = 0.0075, **$p$ = 0.0049, **$p$ = 0.0058, by a t-test. In **v**, **$p$ = 0.0045, *$p$ = 0.0136, **$p$ = 0.0066, *$p$ = 0.0129, by a t-test. Data are presented as mean ± S.D. and are representative of three independent experiments. Source data are provided as a Source Data file.

reduced YTHDF1 expression, may compensate by relying more heavily on YTHDF2 for mRNA decay regulation (Fig. 6p, q).

To confirm the involvement of m⁶A readers in CXCL10 regulation, luciferase reporter assays were performed using wild-type and mutant CXCL10 3′UTR sequences (Fig. S4c, d). In AGS cells, knockdown of YTHDF1 reduced luciferase activity following ALKBH5 knockdown under hypoxia, indicating that YTHDF1 is the primary m⁶A reader mediating CXCL10 regulation in this context (Fig. S4e, f, g, j). In MKN28 cells, however, YTHDF1 knockdown only partially reduced luciferase activity, suggesting the involvement of alternative m⁶A readers in modulating CXCL10 expression (Fig. S4h). Similarly, YTHDF1 knock-down reduced luciferase activity in AGS cells with FTO depletion, while YTHDF2 knockdown increased luciferase activity in MKN28 cells under hypoxia (Fig. S4i, j). These results underscore the crucial role of m⁶A readers in regulating CXCL10 mRNA stability and translation through m⁶A modifications, with distinct contributions from YTHDF1 and YTHDF2 depending on the cellular context.

## Discussion

The discovery of m⁶A modifications has transformed our under-standing of RNA biology, influencing key processes such as splicing[19], translation[20], and RNA stability[17]. Advances in DRS technologies, particularly ONT, have enabled more detailed mapping of m⁶A modifications. However, the complexity of RNA modifications has necessitated the development of computational tools for accurate detection[8,32,36,47–50]. While m6Anet has made pioneering strides in m⁶A detection through multi-instance learning (MIL) on ONT sequencing data[32], its reliance on a Noisy-OR pooling layer and a 20-read threshold introduces limitations. By aggregating multiple reads at each site, m6Anet improves prediction accuracy; however, the 20-read threshold results in the exclusion of lower-coverage loci, potentially limiting the detection of m⁶A sites in low-abundance transcripts[32]. This threshold, while reducing noise, may also over-simplify the complexity of RNA modifications. To address these challenges, pum6a introduces an enhanced MIL framework[33,34] that incorporates both electrical signal and base alignment features, enabling a more comprehensive capture of modification signals. Additionally, the inclusion of a weighted-Noisy-OR probability conversion and an attention-based feature aggregation mechanism offers improved detection sensitivity, particularly at low-read cov-erage sites. Unlike m6Anet, which uses motif-based encoding to capture m⁶A-related patterns, pum6a emphasizes base alignment-derived features, such as base quality, mismatches, and deletions. This approach allows pum6a to detect subtle sequence variations

introduced by m6A modifications, thus enhancing its robustness across a diverse range of contexts[36,48,51,52].

A major challenge in m6A detection is the variability in read coverage across different loci. m6Anet addresses this by filtering out loci with fewer than 20 reads, resulting in reduced noise but also potentially missing important low-abundance modifications. In con-trast, pum6a's weighted-Noisy-OR mechanism allows it to estimate modification probabilities even at low-read coverage sites, broadening its applicability across transcripts with variable expression levels. This adaptability was demonstrated in additional experiments where m6Anet's 20-read threshold was removed, allowing pum6a to out-perform m6Anet across all threshold settings in terms of ROC AUC and PR AUC. An important innovation in pum6a is its attention-based feature aggregation mechanism. While m6Anet employs a traditional MIL approach, pum6a selectively focuses on the most informative reads, which improves the signal-to-noise ratio and detection specifi-city, particularly for m6A sites with low stoichiometry. Furthermore, pum6a's incorporation of positive-unlabeled learning addresses the critical issue of incomplete m6A site labeling in training data[47], enabling it to effectively identify m6A sites even in the presence of incomplete or noisy labels. This makes pum6a particularly effective in detecting m6A modifications in low-abundance transcripts or under-represented biological conditions.

Our application of pum6a to the study of m⁶A demethylases in gastric cancer cells under hypoxia highlights its potential for revealing unrecognized regulatory mechanisms. Notably, pum6a's flexibility in handling read imbalance allowed us to investigate differential m6A modifications across cell lines with variable read coverage. In AGS and MKN28 cells, we observed differential regulation of m6A-modified transcripts, particularly in relation to CXCL10 mRNA stability, indi-cating cell-specific responses to hypoxia. In MKN28 cells, a hetero-zygous deletion of the m⁶A reader YTHDF1 suggests a shift in regulatory dependence towards YTHDF2, which promotes mRNA decay[47]. In AGS cells, where YTHDF1 is fully expressed, CXCL10 mRNA stability is maintained through YTHDF1-mediated recognition of m⁶A sites. This shift in m⁶A reader dependency between the two cell lines underscores the flexibility of m⁶A regulatory networks and the importance of specific m⁶A readers in modulating transcript stability and translation under stress conditions, such as hypoxia. Furthermore, our data suggest that ALKBH5 and FTO target distinct m⁶A sites within CXCL10 mRNA. ALKBH5 appears to regulate modifications outside the 3′UTR, while FTO primarily targets the 3′UTR, providing additional insights into the selective activity of m⁶A demethylases. This differ-ential targeting highlights the complex roles that m⁶A modifications

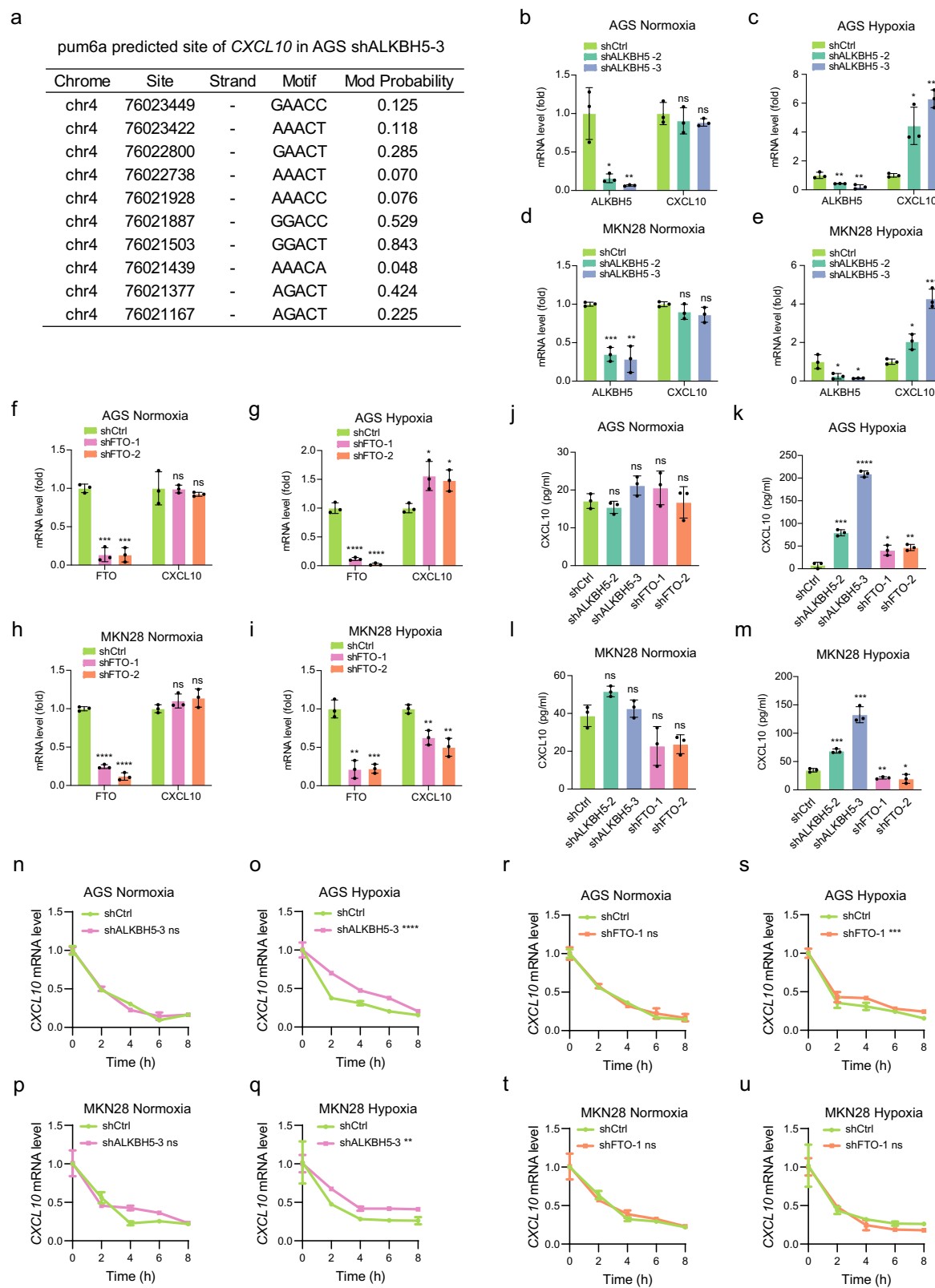

play in regulating gene expression in gastric cancer, particularly in response to hypoxic stress.

While m6Anet provided an important foundation for m⁶A detection from ONT sequencing data, pum6a represents a significant advancement in the field by addressing key limitations inherent in previous methods. Through the integration of both electrical signal and base alignment features, the use of a weighted-Noisy-OR approach

to handle variable read coverage, and an attention-based read selection strategy, pum6a improves detection accuracy and sensitivity. Additionally, its positive-unlabeled learning capability enables pum6a to detect m⁶A sites that may be overlooked by threshold-based methods like m6Anet.

In conclusion, pum6a is not merely an extension of m6Anet but a comprehensive framework that offers more flexibility and nuanced

**Fig. 6 | Hypoxia-induced regulation of CXCL10 expression by m⁶A demethy-
lases in gastric cancer cells. a** Detection of m⁶A modification sites in CXCL10
mRNA in AGS cells following ALKBH5 knockdown, identified by pum6a analysis. **b,
c, d, e** Knockdown of ALKBH5 upregulated CXCL10 mRNA expression in AGS cells
(**b, c**) and MKN28 cells (**d, e**) under hypoxic conditions. In **b**, \*$p = 0.013$,
\*\*$p = 0.0086$, by a t-test. In **c**, \*\*$p = 0.0098$, \*\*$p = 0.0061$, \*$p = 0.0103$, \*\*\*$p = 0.0001$,
by a t-test. In **d**, \*\*\*$p = 0.0003$, \*\*$p = 0.002$, by a t-test. In **e**, \*$p = 0.029$, \*$p = 0.0153$,
\*$p = 0.0129$, \*\*\*$p = 0.0004$, by a t-test. **f, g, h, i** Knockdown of FTO upregulated
CXCL10 mRNA expression in AGS cells (**f, g**) and downregulated CXCL10 expres-
sion in MKN28 cells (**h, i**) under hypoxia. In **f**, \*\*\*$p = 0.0001$, \*\*\*$p = 0.0002$, by a
t-test. In **g**, \*\*\*\*$p < 0.0001$, \*\*\*\*$p < 0.0001$, \*$p = 0.0219$, \*$p = 0.0147$, by a t-test. In
**h**, \*\*\*\*$p < 0.0001$, \*\*\*\*$p < 0.0001$, by a t-test. In **i**, \*\*$p = 0.0012$, \*\*\*$p = 0.0005$,

\*\*$p = 0.0041$, \*\*$p = 0.0026$, by a t-test. **j, k, l, m** ELISA validation of CXCL10 protein
levels in AGS cells (**j, k**) and MKN28 cells (**l, m**) with ALKBH5 or FTO knockdown
under normoxic and hypoxic conditions. In **k**, \*\*\*$p = 0.0002$, \*\*\*\*$p < 0.0001$,
\*$p = 0.012$, \*\*$p = 0.0021$, by a t-test. In **m**, \*\*\*$p = 0.0003$, \*\*$p = 0.0003$, \*\*$p = 0.0031$,
\*$p = 0.0356$, by a t-test. **n, o, p, q** Actinomycin D chase assays showing that ALKBH5
knockdown decelerated CXCL10 mRNA decay rates in AGS cells (**n, o**) and MKN28
cells (**p, q**) under hypoxia. In **o**, \*\*\*\*$p < 0.0001$, by a two-way ANOVA; In
**q**, \*\*$p = 0.0024$, by a two-way ANOVA. **r, s, t, u** FTO Knockdown decreased CXCL10
mRNA decay rates in AGS cells (**r, s**) under hypoxia but promoted mRNA stability in
MKN28 cells (**t, u**). In **s**, \*\*\*$p = 0.0003$, by a two-way ANOVA. Data are presented as
mean ± S.D. and are representative of three independent experiments. Source data
are provided as a Source Data file.

---

detection for m⁶A detection in direct RNA sequencing data. This
advancement broadens opportunities for exploring RNA modifica-
tions and their roles in a wide range of biological processes, particu-
larly in complex and variable environments like cancer and hypoxia.

## Methods

### Pum6a design

Pum6a is an attention-based positive and unlabeled multi-instance
learning framework well-designed for detecting RNA modification
using direct RNA-Seq data.

**MIL with Neural Networks.** In the case of the MIL problem, one aims to
find a model that predicts a value of a target variable, $Y \in \{0, 1\}$, for a
given a bag of instances, $X = \{x_1, \ldots, x_k\}$. Where $k$ represents the
number of instances. We assume that individual labels exist for the
instances within a bag, *i.e.*, $y_1, \ldots, y_k, y_k \in \{0, 1\}$, however they remain
unknown as there is no access to those labels. So, we can re-write the
assumptions of the MIL problem as below:

$$Y = \begin{cases} 0, & if \ \sum_k y_k = 0, \\ 1, & otherwise. \end{cases} \tag{1}$$

These assumptions imply that a MIL model must be permutation-
invariant. We can assume neither ordering nor dependency of instances
within a bag, so the MIL problem can be considered as a specific form of
the Fundamental Theorem of Symmetric Functions with monomials.

In the classical MIL problem, features of instances do not require
further processing. In complex cases like images or text, additional
steps of feature extraction are necessary.

Here, we use a neural network $f_\varphi(.)$ to transform the $k$-th instance
into a low-dimensional embedding, where $\varphi$ represents neural net-
work parameters.

$$h_k = f_\varphi(x_k) \tag{2}$$

Where $x_k$ represent $k$-th instance, and $h_k \in H = \mathbb{R}^M$ represent the low-
dimensional embedding for $k$-th instance.

**MIL pooling.** Classical MIL pooling approaches such as maximum
operator $\forall_{m=1,\ldots,M}: z_m = \max_{k=1,\ldots,K}\{h_{km}\}$ and mean operator
$z = \frac{1}{K}\sum_{k=1}^{K} h_k$, are pre-defined and non-trainable. Here, we propose to
parameterize the neural networks for flexible and adaptive MIL pool-
ing. A weighted average of instances where weights are determined by
neural network was used:

$$z = \sum_{k=1}^{K} a_k h_k \tag{3}$$

Where $h_k$ is the $K$ embedding of $H = \{h_1, \ldots, h_k\}$, and $a_k$ is the
attention-based neural network function to ensure the sum of weights

to 1 to be invariant to the size of a bag:

$$a_k = \frac{\exp\{w^T \tanh(V h_k^T)\}}{\sum_{j=1}^{K} \exp\{w^T \tanh(V h_j^T)\}} \tag{4}$$

Where $w \in \mathbb{R}^{L \times 1}$ and $V \in \mathbb{R}^{L \times M}$ are parameters. $tanh(\cdot)$ is the element-
wise non-linearity hyperbolic tangent to include both negative and
positive values for proper gradient flow.

**Loss function of the MIL learning module.** To simplify the learning
problem, we train the MIL module by optimizing the negative log-
likelihood function, assuming the bag label is distributed according to
the Bernoulli distribution.

$$L_m = -\sum_{k=1}^{N} (y_k \log P_k + (1 - y_k) \log(1 - P_k)) \tag{5}$$

Where: $P_k = \frac{1}{1+e^{-wz}}$, $z$ represent the weighted average of instances and
the $w$ is the parameter.

**Mapping attention scores to probabilities.** Ideally, high attention
weight would be assigned to instances that are likely to have label
$y_k = 1$. That is the key instance would obtain a larger attention score:

$$atts_k = w^T \tanh(V h_k^T) \tag{6}$$

So, we can further map instance attention score to instance
probability using Platt scaling:

$$P_k = \frac{1}{1 + \exp(-\alpha(atts_k) - \beta)} \tag{7}$$

Note that one could apply any transformation function to map
attention score to [0, 1]. We choose Platt scaling as it is widely used in
the literature.

**Transform instance probabilities into bag probabilities.** Taking the
max instance probabilities or an unweighted average of the instance
probabilities to compute the bag probabilities has a clear dis-
advantage. Taking the max instance probability means that the bag
label is based on a single instance and ignores the information in other
samples, which might be inappropriate following the embedding
function mentioned above. Moreover, the mean approach would make
a bag that contains a small number of anomalies seem more normal
than it is. Noisy-OR is a promising alternative, which computes the bag
probabilities of being positive as "one minus the probability that all the
instances are negatives."

$$(Nosiy - OR)\ P(\hat{Y}_B = 1) = 1 - \prod_{j \le k}(1 - f(x_j))$$

The standard Noisy-OR is clearly inappropriate in the case of ONT data, as different sites may contain various numbers of reads. The standard noisy-OR approach produces bag probabilities of being positive that converge exponentially to 1 for $k \rightarrow +\infty$. In this research, weighted Noisy-OR is used, which gives higher weight to the instances with the highest and the lowest positive probabilities.

$$\text{(Weighted nosiy} - \text{OR)} \; P\left(\hat{Y}_B = 1\right) = 1 - \prod_{x_j \epsilon B} \left(1 - f\left(x_j\right)\right)^{w_j} \quad (8)$$

Where $w_j$ is the weight for $x_j$, and is calculated as follow:

**First.** we rank the positive instance probabilities in ascending order using a ranking map $p_f : \mathbb{R}^k \rightarrow \{0, \ldots, k-1\}$ and normalize the rankings to [0, 1] by dividing them by $k$-1.

$$p_f\left(x_j\right) = r \epsilon \{0, \ldots, k-1\} \Longleftrightarrow \left\{ \begin{array}{l} |x \epsilon \{x_1, \ldots, x_k\} : f(x) < r| = r \\ |x \epsilon \{x_1, \ldots, x_k\} : f(x) > r| = k - r - 1 \end{array} \right. \quad (9)$$

**Second.** we introduce a weight function $S : [0, 1] \longrightarrow \mathbb{R}$ that gives high weights to both high and low rankings.

$$S\left(\frac{p_f\left(x_j\right)}{k-1}\right) = N_0\left(\frac{p_f\left(x_j\right)}{k-1}\right) + N_1\left(\frac{p_f\left(x_j\right)}{k-1}\right) \quad (10)$$

Where $N_\alpha$ is the Gaussian density function with mean $\alpha$ and standard deviation 0.1. Here, we define $S$ as $N_0 + N_1$ to make such function to have two peaks. (0 for low rankings and 1 for high rankings) and be flat (almost null) in between.

**Third.** we apply the function $S$ to each instance's ranking and normalize them within bag.

$$w_j = S\left(\frac{p_f\left(x_j\right)}{k-1}\right) / \sum_{q<k} S\left(\frac{p_f\left(x_q\right)}{k-1}\right) \quad (11)$$

**Selecting the *R* reliable negatives.** In the positive and unlabeled task, learning from only positive bag labels would make the model overfit towards the positive class. Here, we selected $|R|$ reliable negatives negative bags among $B$ with the lowest positive probability. We selected $|R|=|P|$, to transform the problem into a classification task with balanced classes.

**Loss function of the positive and unlabeled learning module.** Similarly, as we assume that labels follow a Bernoulli distribution with parameter $F(B)$, we build the loss function for this module as:

$$L_p = -\log\left(\prod_{B \in P} F(B) \prod_{B \in R} (1 - F(B))\right) + \lambda\left(\alpha^2 + \beta^2\right) \quad (12)$$

Where $\left(\alpha^2 + \beta^2\right)$ avoid overfitting, and we derive pum6a's loss function as

$$L = L_m + L_p \quad (13)$$

## Pum6a for modification detection from nanopore direct RNA sequence data

**Base-calling, re-squiggle, and mapping.** Training and evaluation data of the pum6a model were obtained from refs. We downloaded a single replicate of the HEK293T cell lines (replicate 1) for model training. We use another independent single replicate of the HEK293T cell lines (replicate 4) for model evaluation. We further tested the model performance in other species (mouse embryonic stem cells) and constructed data (Curlcake). Reads were locally base-called using Guppy 6.5.7. MINES [11] corrected the raw signal through Tombo re-squiggle function. After base-calling, we performed re-squiggle with Tombo (v1.5) to correct the raw base-calling sequence and assign the corrected base to the raw signal segment. We finally mapped the sequences to the genome using minimap2 with the settings -ax map-ont.

**Feature extraction.** We searched for the RRACH motif and extracted the signal features and the sequence features per site. As for the signal feature, we extracted the median, standard deviation, mean, and number of Nanopore signals form each RRACH motif. As for sequence features, we use sam2tsv from jvarkit to convert BAM alignment files to tab delimited. We extract mean per-base quality, mismatch frequency, insertion frequency, and deletion frequency for each RRACH motif.

**Model training and evaluation.** Modified position of HEK293T from miCLIP, m6ACE-seq, and GLORI were obtained from ref. We combined CIMS, CITS, m6ACE-seq and GLORI libraries as ground truths for model training and evaluation. We train the model with HEK293T data and further evaluate the model in a mouse embryo sample. We downloaded miCLIP1 and miCLIP2 libraries of mES from the ref and combined them as ground truth labels for model evaluation.

## Comparison with other models for m⁶A detection

**Tombo.** We ran Tombo v.1.5.1 from https://github.com/nanoporetech/tombo. After re-squiggle reads, we assign raw current signals to each base. We then use a de novo non-canonical base method with the "detect_modifications de_novo" command for modification detection.

**EpiNano.** We ran EpiNano 1.2 from https://github.com/enovoa/EpiNano. We collected the mean quality, mismatch, insertion, and deletion frequency of each base of the RRACH motif. Moreover, we use the Support Vector Machine models EpiNano offers for m⁶A detection.

**MINES.** We ran MINES from https://github.com/YeoLab/MINES. Mines trained the random forest model for m⁶A detection using the coverage and fraction-modified values calculated by Tombo. As MINES only use for AGACT and GGACH motifs detection. We modified the code to output the modification probability of all sites in all RRACH motifs.

**Nanom6A.** We ran nanom6A from https://github.com/gaoyubang/nanom6A. Nanomo6a trained the XGBoost model for m⁶A detection using the median, mean, standard deviation and dwell time features from normalized raw signals.

**ELIGOS.** We ran ELIGOS from https://gitlab.com/piroonj/eligos2. We used the "rna_mod" of ELIGOS for the identification of RNA modifications compared to an rBEM + 2 model.

**M6anet.** We ran m6anet from https://github.com/GoekeLab/m6anet. We used Nanopolish (0.14.1) to generate the input required by m6anet.

## Gene level counts of DRS data

In long-read mode, we used featureCounts version 1.6.3 for each nanopore DRS replicate Gene level counts[53].

## Enrichment analysis

We used web server DAVID for functional enrichment analysis[54].

## Cell culture and treatments

The cells were maintained at 37 °C in a humidified incubator with 5% CO2. For the hypoxic culture, the cells were cultured in a low oxygen incubator with a gas mixture containing 1% O2, 5% CO2, and 94% N2.

HEK-293T cells (ATCC, CRL-3216, USA) were cultured in DMEM (Gibco), while AGS (ATCC, CRL-1739, USA) and MKN28 (BNCC, 338339, China) cells were cultured in RPMI (Gibco). The culture media were supplemented with 10% FBS (Gibco) and 1% penicillin and streptomycin (Gibco).

## Plasmid construction, lentiviral production and tumor cell infection

Stable knockdowns of FTO, ALKBH5, YTHDF1 and YTHDF2 were generated using lentiviral-based shRNA. For each knockdown, 10 µg of PLKO.1 plasmid (PLKO.1 puro #8453, PLKO.1 hygro #24150, addgene) containing shRNA targeting the gene of interest, along with 5.6 µg of PAX2 (#12260, addgene) and 3.5 µg of pMD2.G (#12259, addgene) packaging vectors, were co-transfected into HEK-293T cells in 100 mm cell-culture dishes. The shRNA sequences used for FTO knockdown were shFTO (1, CGGTTCACAACCTCGGTTTAG; 2, TCACCAAGGAGACT GCTATTT), for ALKBH5 knockdown were shALKBH5 (1, GAAAGGC TGTTGGCATCAATA; 2, CCACCCAGCTATGCTTCAGAT; 3, CCTCAG-GAAGACAAGATTAGA), for YTHDF1 knockdown were shYTHDF1 (CCCTACCTGTCCAGCTATTAC) and for YTHDF2 knockdown were shYTHDF2 (TCTGGATATAGTAGCAATTAT). After 48 and 72 h, the virus-containing supernatant was collected and filtered through a 0.22 µm PES Syringe Filter (Thermo Fisher) before infecting the target cells with 8 µg/mL Polybrene (TR-1003, Sigma). The cells were then screened with puromycin (2 µg/mL) at the corresponding concentration for 3-5 days to obtain stable knockdown cells. The FTO cDNA (NM_001080432) and ALKBH5 cDNA (NM_017758) were cloned into pCDH-puro lentiviral vector (CD510B-1, System Biosciences).

## Poly(A) RNA isolation

100 µg aliquots of total RNA were diluted in 100 µl of nuclease-free water and subjected to poly(A) selection using Magnosphere MS150/Oligo(dT) Beads (Lexogen, Poly(A) RNA Selection Kit, cat. no. 039.100). The eluted poly(A) RNA was measured using a Qubit fluorometer, and stored at -80 °C for MinION native RNA sequencing.

## MinION native RNA sequencing

For the preparation of RNA libraries, 500 ng of poly(A) RNA were used, and the Direct RNA Sequencing Kit (SQK-RNA002) was employed for nanopore direct RNA sequencing. The ligation of the RT Adapter and RNA Adapter was carried out using T4 DNA Ligase (M0202, NEB) and SuperScript III Reverse Transcriptase (18080044, Thermo Fisher Scientific). The RNA library was purified using Agencourt RNAClean XP beads. SpotON flow cells were primed and loaded according to the kit protocol for RNA sequencing on the MinION platform.

## Proliferation and viability assays

Cell viability was assessed using the Cell Counting Kit-8 (CCK-8 assay, DOJINDO) following the manufacturer's instructions. The proliferative capacity of AGS and MKN28 cells was measured by cell counting and the BeyoClick™ EdU Cell Proliferation Kit with Alexa Fluor 555 (C0075L, Beyotime).

## RNA isolation and quantitative real-time PCR

Total RNA was extracted from tumor cells using TRIzol reagent (15596018, ThermoFisher) following the manufacturer's instructions, and cDNA was synthesized using the PrimeScript RT reagent Kit (RR036A, Takara). RT-qPCR was performed on the 7500 apparatus (Applied Biosystems) using SYBR-Green Master mix (RR820B, Takara). The following primer sequences were used: GAPDH (Forward, GTCTCCTCTGACTTCAA CAGCG; Reverse, ACCACCCTGTTGCTGTAGCCAA), CXCL10 (Forward, GTGGCATTCAAGGAGTACCTC; Reverse, TGATGGCCTTCGATTCTGGAT T), YTHDF1 (Forward, CAAGCACACAACCTCCATCTTCG; Reverse, GTAA GAAACTGGTTCGCCCTCAT) and YTHDF2 (Forward, TAGCCAGCTACAA GCACACCAC; Reverse, CAACCGTTGCTGCAGTCTGTGT).

## Western blotting

Protein samples were extracted using RIPA lysis buffer containing a protease inhibitor cocktail (Roche) and the protein concentration was quantified using the BCA method. A total of 30 µg of protein was loaded and electrophoresed on 12% SDS–polyacrylamide gels, transferred onto PVDF membranes (Invitrogen), and subjected to western blot analysis. Antibodies were diluted in 5% (wt/vol) nonfat dry milk in PBS containing 0.1% Tween-20. The primary antibodies used were rabbit monoclonal anti-ALKBH5 (1:1000, A11684, ABclonal), rabbit monoclonal anti-FTO (1:1000, A3861, ABclonal), and anti-GAPDH (1:1000, 5147, Cell Signaling Technologies).

## ATP and NAD + /NADH assay

The levels of ATP and NAD + /NADH were measured using the ATP Assay Kit (S0026, Beyotime) and NAD + /NADH Assay Kit with WST-8 (S0175, Beyotime), respectively. The protein samples were quantified using the BCA protein assay kit (P0011, Beyotime) to determine the ATP or NAD + /NADH levels per microgram of protein.

## RNA stability

The cells were treated with actinomycin D (HY-17559-5 mg, MCE) at a final concentration of 5 µg/mL for the indicated time period, after which they were collected. Real-time PCR was then performed to determine the relative abundance of each mRNA.

## ELISA

Supernatants from AGS, MKN28, and their corresponding stable knockdown cells for FTO or ALKBH5 were used to measure the concentrations of CXCL10. The Human CXCL10 ELISA Kit (EHC157.96, Neobioscience) were used according to the manufacturer's instructions to quantify the concentrations of CXCL10.

## Luciferase reporter assays and mutagenesis analysis

The 3'UTR sequence of CXCL10 was amplified by PCR from AGS cell genomic DNA and subsequently subcloned into the dual-luciferase vector pmiGLO (C838A, Promega). Predicted m6A recognition sites within the 3'UTR were identified through the pum6A analysis. Site-directed mutagenesis, altering adenine (A) to thymine (T), was performed using the QuikChange II Site-Directed Mutagenesis Kit (200523, Agilent). Luciferase activity was measured using the dual-luciferase reporter assay kit (RG028, Beyotime) in the GM2000 luminometer (Promega). All experiments were performed in triplicate, and firefly luciferase activity was normalized to Renilla luciferase activity to account for variations in transfection efficiency.

## Statistics and reproducibility

No statistical method was used to predetermine the sample size. No data were excluded from the analyses. The experiments were not randomized, and the investigators were not blinded to allocation during experiments and outcome assessment. Data are presented as the mean ± standard deviation (S.D.). Statistical analyses were performed using GraphPad Prism 10.0 software. Statistical differences between the indicated groups were assessed using a two-tailed Student's t-test or two-way ANOVA. A $p$ value of less than 0.05 was considered statistically significant. Statistical significance is indicated as follows: $p < 0.05$; *$p < 0.01$; **$p < 0.001$. Biological replicates and the number of independent experiments are stated in the figure legends. All experiments presented as representative micrographs or gels were repeated at least 3 times with similar results.

## Reporting summary

Further information on research design is available in the Nature Portfolio Reporting Summary linked to this article.

## Data availability

The HEK293T cell lines data generated in this study have been deposited in the European Nucleotide Archive (ENA) under accession code PRJEB44348. The mouse embryo data used in this study are available in the Gene Expression Omnibus (GEO) database under accession code GSE195618. The constructed data were obtained from the GEO database under accession code GSE124309. The benchmark anomaly datasets used in this study are publicly accessible at https://github.com/Minqi824/ADBench/tree/main/adbench/datasets/Classical. The data generated in this study are provided in the Supplementary Information/Source Data file. Source data are provided with this paper.

## Code availability

The source code for pum6a is publicly available at https://github.com/liuchuwei/pum6a and the doi for the code is https://doi.org/10.5281/zenodo.14279615.

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

## Acknowledgements

This study was partly supported by the National Key Research and Development Plan 2022YFC2402900 (ZW), 2022YFC3401000 (WH); National Natural Science Foundation of China 82473938 (GW), 82122069 (GW), 82073869 (GW), 82022037 (WH), U22A20315 (ZW); Guangdong Basic and Applied Basic Research Foundation 2021B1515020004 (GW), 2021B1515230009 (WH); Fundamental Research Funds for the Central Universities 23yxqntd001 (GW), 24ykzy003 (GW); Guangzhou Science and Technology Planning Program 2024A04J6479 (GW); and Shenzhen Medical Research Special Fund Project D2403006 (GW).

## Author contributions

G.W. and C.L. conceived the idea and designed the experiments. C.L., H.L., A.H.W., M.X., L.S., Y.Y., S.Y., Y.D., R.L. and J.F. performed the most experiments and analyzed data; C.L. performed the bioinformatics analysis. Z.W., W.H. and G.W. provided administration and supervision. G.W. and C.L. wrote the manuscript. All authors were involved in the final approval of the submitted and published versions.

## Competing interests

The authors declare no competing interests.
