## [Transparent Peer Review file · Nature Communications]

Decoding the m6A Epitranscriptomic Landscape for Biotechnological Applications Using A Direct RNA Sequencing Approach

Corresponding Author: Professor Guohui Wan

Version 0:

Reviewer comments:

Reviewer #2

(Remarks to the Author)

The paper presents a computational framework, pum6a, for identifying m6A modifications in RNA using direct RNA sequencing data. Pum6a uses an attention-based positive and unlabeled multi-instance learning strategy. The authors claim that puma6a shows superior performance across different cell lines and species. It has the potential for understanding m6A dynamics in diseases such as gastric cancer under hypoxic stress.

My major concern with this paper is the lack of clarity regarding the relationship between pum6a and another multiple-instance learning framework, m6Anet. It's clear that pum6a is heavily influenced by m6anet. However, the text does not explicitly state this, and the authors have not clearly outlined the improvements or provided evidence of their impact on the results. While it's common to use someone else's code and methodology as a starting point, it's essential to clearly distinguish the original work and provide a detailed assessment of the improvements.

I have found the following similarities:

1. Introduction - some paragraphs look just paraphrased
2. Source code - the license is still from Goke lab (authors of m6anet)
3. The similarity between Figure 3a in pum6a paper and Figure 1a in m6anet paper. Even logos are similar.
4. Equation 5—The same equation is used in the m6anet paper, and this equation is evidently wrong. Interestingly, the authors in the m6anet paper discuss minimising cross-entropy loss, while the pum6a authors use another term: maximum likelihood estimation of Bernoulli distribution. However, in both cases, the equation is exactly the same and incorrect.
5. Precision recall curves for these two methods in pum6a paper are often almost identical.

I argue that the authors first need to do a detailed analysis and convince the readers that their improvement over m6anet is significant and give authors of m6anet proper credit.

(Remarks on code availability)

I didn't make a detailed analysis of the code.

Reviewer #3

(Remarks to the Author)

1. While the authors claimed that they validated that “under hypoxia, ALKBH5 expression increased, whereas FTO's decreased”, no data presented in Figure 5 shows this effect on the protein level, and the gene expression data in Figure 5c did show it for ALKBH but not for FTO (the difference between hypoxia and normoxia was not significant). Western blot data should be presented for the effect of hypoxia on the expression of ALKBH5 and FTO. And the author should explain the no difference between hypoxia and normoxia for the gene expression of FTO.

2. ALKBH5 and FTO are both demethylase, but one supposedly goes up and the other goes down under hypoxia, this means that they are supposed to play opposite roles in adaptation of cells to hypoxia. However, their knock down (reversing

the effect of hypoxia for the first and mimicking the effect of hypoxia for the second) resulted in the same biological outcome (inhibition of cell proliferation) under hypoxia. How is this possible? If anything the experiments should have been performed with overexpression of FTO.

3. The results section claims that “AGS cells experienced a marked reduction in proliferation under hypoxic conditions”, while this is correct for cell count (in Figure 5d-normoxia with maximal growth of about 1000000 cells, compared to Figure 5e-hypoxia with maximal growth of about 500000), this is not the case for data obtained with “growth rate (Figure 5f-g) measurements” which shows identical proliferation rate for both hypoxia and normoxia, with maximal growth of untreated of 1.5 in both cases.

4. It is hard to make conclusions from Figures 5h-l; first the images seem to be presented with different brightness/contrast, this is obvious for example shALKBH5-2 vs. shFTO-1, and in both Figure 5h and Figure 5i. These images must be re-taken or presented with the same intensity/contrast conditions.

5. The data in Figure 5h-l is qualitative, and a more quantitative approach should be presented such as mean fluorescence intensity of EDU of all cells in each frame, and/or the percentile of EDU-high intensity cells from all Hoechst+ cells.

6. “Both aerobic and anaerobic metabolism rely on the electron carrying properties of pyridine nucleotides to regulate energy production. The intracellular NAD⁺/NADH ratio controls the rate of ATP synthesis [PMID: 32258851]”. The data in Figure 5o-r show that the wt cells followed the expected effect; in which hypoxia reduced ATP production (from 0.25 in Figure 5o to 0.18 in Figure 5p), and similarly reduced the NAD⁺/NADH ratio (from 50 in Figure 5q to 4 in Figure 5r). Similarly, as expected, compared to wild-type cells, knockdown cell lines had lower ATP production in both hypoxia (Figure 5p) and normoxia (Figure 5o), as well as reduced NAD⁺/NADH ratio in normoxia (Figure 5q). However, the opposite pattern was observed in NAD⁺/NADH ratio in hypoxia (Figure 5r), where knockdown cells showed higher NAD⁺/NADH ratio compared to wild-type, these results contradict the ATP production results (Figure 5p), as well as the cell proliferation results (Figure 5e).

7. While the effect of hypoxia was clear reduction of ATP in all wild-type and knockdown cells (comparing each column in Figure 5o to its counterpart in Figure 5p), the effect of hypoxia on NAD⁺/NADH ratio was variable in the different cells (comparing each column in Figure 5q to its counterpart in Figure 5r). In wild type cells (green columns) it was dramatically reduced, in shALKBH5-2 (turquoise columns) it was dramatically increased, in shALKBH5-3 it was not changed, in shFTO-1 it was increased, in shFTO-2 it was not changed or a slight decrease. These inconsistencies must be addressed.

8.(Minor Comment): the Y-axis scales in Figures 5 d&e, o&p, q&r, should be unified for better comparison.

9. While knock down of ALKBH5 and FTO had no effect on CXCL10 protein expression in normoxia, neither in AGS (Figure 6f) nor in MKN28 cells (Figure 6h), it did change the expression in hypoxia (Figures 6g&i). While the knockdown of ALKBH5 consistently increased the expression of CXCL10 in both cell lines (Figure 5g&i), the effect of knockdown of FTO was inconsistent, showing an increase in AGS cell line (Figure 5g), and a decrease in MKN28 cell line (Figure 6i). The authors should address these inconsistencies.

10.While Figures 6b-e show mRNA expression that aligns with the protein expression data for the ASLBH5 knockdown, no such data was provided for the FTO knockdown. We think this is important data to show a complete story.

11. Based on the short-comes of the biological data, it is hard to make the conclusion of “by applying pum6a to m6A modification studies, we have elucidated the significant roles of the m6A demethylases FTO and ALKBH5 in the adaptation of gastric cancer to hypoxic conditions”.

(Remarks on code availability)

Reviewer #4

(Remarks to the Author)

(Remarks on code availability)

Reviewer #5

(Remarks to the Author)

(Remarks on code availability)

Version 1:

Reviewer comments:

Reviewer #2

(Remarks to the Author)

The authors have addressed some of my initial concerns, revising the text accordingly and appropriately crediting m6Anet. They also made efforts to adjust the content to minimize similarities between papers, addressing common errors and ensuring sufficient textual differentiation. However, after a thorough re-evaluation of the article and the m6Anet paper, I still view the methodological contributions as largely incremental. Below, I outline my specific concerns, taking the authors' responses into account.

#Origin and Innovation.

Authors claim: "m6Anet is based primarily on extracting electrical signal features from nanopore sequencing for m6A detection (1-5). In contrast, pum6a goes further by integrating both electrical signal features and base alignment information, allowing for a more comprehensive feature capture"

From the m6Anet paper, it is clear that they use both signal feature and sequence signal (ie. Figure 1a)

Algorithmic Differences

- 1.Feature Extraction. Contrary to what authors claim, both methods use electrical signal information and sequence
- 2.Handling Read Number Imbalance. Authors claim their method does not have a hard threshold of 20 reads like m6Anet. I agree. It would be interesting to compare results in the case when there is no threshold in m6Anet
- 3.Adaptability to Positive and Unlabeled Data. Authors claim that "Pum6a's use of an attention-based feature aggregation mechanism enables it to extract key information effectively from complex datasets".

The m6Anet paper authors claim: "We have evaluated alternative pooling layers, such as the Attention and gated Attention-based pooling, but have not found any statistically significant improvement in the performance of m6Anet compared to the noisy-OR pooling layer for m6A detection."

An ablation study is needed to prove that this provides a significant increase in performance.

Advantages and Performance

Figures 3g, 3h, 4g, and 4g compare pum6a and m6Anet. Apart from 3h, the results are the same, with negligible differences.

As mentioned above, I agree that m6Anet is limited to at least 20 reads, but it would be interesting to see the performance without that threshold. Also, if there is any other reason why pum6a reports more sites, it should be clearly stated.

(Remarks on code availability)

I have installed the pum6a code and found no difficulties during the installation. It would be important that authors prepare a toy example with necessary data (or link to them) so that additional tests and verifications can be made.

Reviewer #3

(Remarks to the Author)

The authors have fulfilled all the revisions needed.

(Remarks on code availability)

Reviewer #4

(Remarks to the Author)

(Remarks on code availability)

Reviewer #5

(Remarks to the Author)

(Remarks on code availability)

Version 2:

Reviewer comments:

Reviewer #2

(Remarks to the Author)

I want to thank the authors who have made all my suggested experiments.

Taking into account Figures 3 and 4 and Supplementary Figure 1, it looks like there are some improvements in performances in comparison to 6maNet. However, I am still not convinced that the proposed changes and differences in performance are significant enough for someone to use pum6A instead of 6maNet.

(Remarks on code availability)

The authors have uploaded some toy examples. Unfortunately, there are no instructions on how to test them quickly.

Response to Reviewers (Manuscript ID: NCOMMS-24-24836-T)

We sincerely thank the reviewers for their valuable and insightful comments, which have significantly contributed to the improvement of our manuscript. In response to their feedback, we have conducted a comprehensive comparative analysis between m6Anet and our pum6a framework, focusing on the algorithmic differences and the specific innovations introduced by pum6a. Additionally, we have conducted a series of supplementary experiments to further elucidate the roles of ALKBH5 and FTO in mediating the cellular response to hypoxia via m6A modification. These experiments include (1) assessments of cell proliferation following ALKBH5 and FTO overexpression in gastric cancer cells; (2) validation of energy homeostasis disruption upon m6A demethylase depletion; (3) evaluation of m6A regulation on CXCL10 through 3'UTR luciferase reporter assays, and (4) investigation of the role of m6A readers in cell-specific CXCL10 RNA metabolism. These new findings reinforce the robustness of pum6a in detecting m6A modifications, particularly in low-abundance transcripts and under-represented biological conditions. They also deepen our understanding of the functional significance of ALKBH5 and FTO in the hypoxic response of gastric cancer cells. We have incorporated the new data into the manuscript, revised the figures accordingly, and highlighted the changes in blue. Our responses to the reviewers' comments are detailed point-by-point below.

Reviewer #2:

1. *"My major concern with this paper is the lack of clarity regarding the relationship between pum6a and another multiple-instance learning framework, m6Anet. It's clear that pum6a is heavily influenced by m6anet. However, the text does not explicitly state this, and the authors have not clearly outlined the improvements or provided evidence of their impact on the results. While it's common to use someone else's code and methodology as a starting point, it's essential to clearly distinguish the original work and provide a detailed assessment of the improvements."*

We appreciate reviewer #2's concern regarding the relationship between pum6a and m6Anet and the need for clarity in distinguishing the two frameworks. While pum6a is influenced by m6Anet, it represents a distinct development with significant advancements. Below, we outline the origin, uniqueness, and differences between pum6a and m6Anet (this comparison is also included in discussion in our revised manuscript):

Origin and Innovation: Our research focuses on understanding mRNA modifications in oncology using nanopore sequencing technology, an area in which various algorithms like m6Anet have inspired us. m6Anet is based primarily on extracting electrical signal features from nanopore sequencing for m6A detection (1-5). In contrast, pum6a goes further by integrating both electrical signal features and base alignment information, allowing for a more comprehensive feature capture. This integration enhances the sensitivity and specificity of m6A detection, addressing limitations identified in m6Anet and other algorithms (6-9).

Algorithmic Differences: Although both pum6a and m6Anet aim to develop high-precision supervised learning algorithms for m6A modification detection using experimental labels, they differ significantly in their algorithmic approach:

1). Feature Extraction: Pum6a incorporates both electrical signal and base alignment features, providing a more holistic view of the data, whereas m6Anet focuses solely on electrical signal features.

2). Handling Read Number Imbalance: Unlike m6Anet, which filters out gene loci with fewer than 20 reads to reduce noise, pum6a uses a weighted-Noisy OR model to manage the variability in read coverage across gene loci, enabling broader identification without excluding low-coverage loci.

3). Adaptability to Positive and Unlabeled Data: Pum6a is specifically designed to handle positive and unlabeled multi-instance data, whereas m6Anet deals with traditional multi-instance data. Pum6a's use of an attention-based feature aggregation mechanism enables it to extract key information effectively from complex datasets.

4). Algorithmic Inspiration: Pum6a is primarily inspired by the Puma algorithm (10), which integrates positive and unlabeled learning into a unified framework, and the ATMIL algorithm (11), which applies an attention mechanism for complex feature data. We combined these algorithms to address the unique challenges of third-generation sequencing data, distinguishing pum6a fundamentally from m6Anet.

Advantages and Performance: Pum6a's design offers several advantages over m6Anet:

1). Enhanced Sensitivity and Specificity: By utilizing both electrical signal and base alignment features, pum6a provides more depth in information mining, identifying approximately 5 to 10 times more m6A modification sites than m6Anet.

2). Robustness: The weighted-Noisy OR mechanism enables pum6a to maintain high detection stability even with significant variations in gene locus read counts.

3). Versatility: The attention-based feature extraction allows pum6a to handle various complex datasets, demonstrating its applicability and robustness across different biological contexts.

In summary, while pum6a and m6Anet share a common goal and utilize experimental labels for training (12), pum6a introduces significant innovations in feature extraction, algorithm design, and handling of positive and unlabeled data. These differences result in pum6a's enhanced performance and broadened applicability in m6A modification detection.

Reference:

1. Gao Y. *et al.* Quantitative profiling of N6-methyladenosine at single-base resolution in stem-differentiating xylem of *Populus trichocarpa* using Nanopore direct RNA sequencing. *Genome Biol.* 2021 Jan 7;22(1):22. doi: 10.1186/s13059-020-02241-7.
2. Hendra C. *et al.* Detection of m6A from direct RNA sequencing using a multiple instance learning framework. *Nat Methods.* 2022 Dec;19(12):1590-1598. doi: 10.1038/s41592-022-01666-1.
3. Leger A. *et al.* RNA modifications detection by comparative Nanopore direct RNA sequencing. *Nat Commun.* 2021 Dec 10;12(1):7198. doi: 10.1038/s41467-021-27393-3.
4. Lorenz D. *et al.* Direct RNA sequencing enables m6A detection in endogenous transcript isoforms at base-specific resolution. *RNA.* 2020 Jan;26(1):19-28. doi: 10.1261/rna.072785.119. Epub 2019 Oct 17.
5. Pratanwanich P. *et al.* Identification of differential RNA modifications from nanopore direct RNA sequencing with xPore. *Nat Biotechnol.* 2021 Nov;39(11):1394-1402. doi: 10.1038/s41587-021-00949-w.

6. Jenjaroenpun P. et al. Decoding the epitranscriptional landscape from native RNA sequences. *Nucleic Acids Res.* 2021 Jan 25;49(2):e7. doi: 10.1093/nar/gkaa620.
7. Liu H. et al. Accurate detection of m6A RNA modifications in native RNA sequences. *Nat Commun.* 2019 Sep 9;10(1):4079. doi: 10.1038/s41467-019-11713-9.
8. Parker M. et al. Nanopore direct RNA sequencing maps the complexity of Arabidopsis mRNA processing and m6A modification. *Elife.* 2020 Jan 14;9:e49658. doi: 10.7554/eLife.49658.
9. Price A. et al. Direct RNA sequencing reveals m6A modifications on adenovirus RNA are necessary for efficient splicing. *Nat Commun.* 2020 Nov 26;11(1):6016. doi: 10.1038/s41467-020-19787-6.
10. Perini, L. et al. in *Proceedings of the 29th ACM SIGKDD Conference on Knowledge Discovery and Data Mining 1897–1906* (Association for Computing Machinery, Long Beach, CA, USA, 2023).
11. Ilse M. et al. Attention-based Deep Multiple Instance Learning. *Proceedings of the 35th International Conference on Machine Learning*, PMLR 80:2127-2136, 2018.
12. Zhong Z. et al. Systematic comparison of tools used for m6A mapping from nanopore direct RNA sequencing. *Nat Commun.* 2023 Apr 5;14(1):1906. doi: 10.1038/s41467-023-37596-5.

2. *“Introduction - some paragraphs look just paraphrased.”*

We thank Reviewer #2 for raising this concern. To address the issue, we performed a thorough check using Copyleaks, a professional plagiarism detection tool, which confirmed a similarity score of 0%. We understand that in a specialized field like third-generation sequencing, especially with rapidly evolving topics such as m6A modifications, certain sections of the introduction may appear similar due to the use of common terminology and foundational concepts that are widely accepted in the literature (1, 2).

However, to ensure clarity and originality, we have rewritten the introduction to provide a unique perspective on the current state of research. The revised content, which includes a more focused narrative on the significance and innovation of our work, has been highlighted in blue in the manuscript.

Figure Redacted

Reference:

1. Zhong, Z. D. et al. Systematic comparison of tools used for m(6)A mapping from nanopore direct RNA sequencing. *Nat Commun* 14, 1906 (2023).
2. Liu, C. et al. Absolute quantification of single-base m6A methylation in the mammalian transcriptome using GLORI. *Nature Biotechnology* 41, 355-366 (2023).

3. *“Source code - the license is still from Goke lab (authors of m6anet).”*

We appreciate reviewer#2 bringing this issue to our attention. We have addressed the concern regarding the source code's license. The updated code now adheres fully to the MIT License. The

license text explicitly grants permission for unrestricted use, modification, and distribution, as follows: "Permission is hereby granted, free of charge, to any person obtaining a copy of this software and associated documentation files (the 'Software'), to deal in the Software without restriction, including without limitation the rights to use, copy, modify, merge, publish, distribute, sublicense, and/or sell copies of the Software, and to permit persons to whom the Software is furnished to do so." This ensures that our code is compliant with open-source standards and clearly distinguishes it from any prior work.

4. *"The similarity between Figure 3a in pum6a paper and Figure 1a in m6Anet paper. Even logos are similar."*

We acknowledge the reviewer's observation regarding the resemblance between our Figure 3a and Figure 1a in the m6Anet paper. However, we would like to clarify that Figure 3a in our manuscript is a conceptual diagram designed to illustrate the overall framework of how pum6a extracts features and constructs models in the context of third-generation sequencing. This figure is not intended to showcase experimental data but rather to provide a clear schematic representation of the pum6a framework. It is divided into four distinct parts, each highlighting unique aspects of our approach.

Firstly, the diagram emphasizes pum6a's innovation in utilizing both sequence signals generated by nanopore full-length sequencing and base alignment quality, setting it apart from m6Anet, which primarily relies on a combination of electrical signal features and sequence features. Secondly, it illustrates how pum6a addresses the challenge of positive and unlabeled multi-instance data by treating samples not explicitly detected in experiments as potential unlabeled data, a methodology fundamentally different from m6Anet's traditional approach. Lastly, the figure incorporates the fully connected layer, a standard component in neural networks, as part of the model architecture.

Regarding the logo similarity, we understand the concern may arise due to the inclusion of the keyword "m6A." However, the pum6a logo design is inspired by a puma and integrates the concepts of "pu" (representing positive and unlabeled) and "ma" (symbolizing multi-instance), with the number "6" placed between "m" and "a" to indicate the algorithm's specific design for m6A detection in third-generation sequencing. We believe this logo is original and distinct, serving as a unique mark of the pum6a brand with its symbolic expression, smooth lines, and aesthetic appeal.

To further address this concern, we have revised Figure 3a to more distinctly differentiate its design and layout from that of the m6Anet paper while ensuring it accurately represents the core aspects of our framework. The revised figure has been included in the updated manuscript.

Figure Redacted

Figure 1a (the m6Anet paper)

Revised Figure 3a (our pum6A manuscript)

5. *“Equation 5—The same equation is used in the m6Anet paper, and this equation is evidently wrong. Interestingly, the authors in the m6Anet paper discuss minimising cross-entropy loss, while the pum6a authors use another term: maximum likelihood estimation of Bernoulli distribution. However, in both cases, the equation is exactly the same and incorrect.”*

We appreciate the reviewer pointing out the similarity between our Equation 5 and the one used in the m6Anet paper. This resemblance arises from both addressing similar challenges in the domain of m6A detection. However, we would like to clarify that our approach in pum6a is primarily derived from the Puma framework (1) and has been specifically enhanced using the ATMIL (Attention-based Multi-Instance Learning) approach to better handle complex feature areas such as third-generation sequencing and image recognition (2). While there is an overlap in the application domain between pum6a and m6Anet, our method is distinct in its adaptation to address challenges like read number imbalance through a weighted-Noisy OR mechanism and the issue of positive and unlabeled samples.

Regarding the concerns about Equation 5, we recognize that it was not correctly formulated in our original submission. We have revised Equation 5 as follows to accurately represent the maximum likelihood estimation of the Bernoulli distribution:

$$L_m = - \sum_{k=1}^N (y_k \log P_k + (1 - y_k) \log(1 - P_k)) \quad (5)$$

Where: $P_k = \frac{1}{1 + e^{-wz}}$, z represent the weighted average of instances and the w is the parameter.

Reference:

1. Perini, L. et al. in Proceedings of the 29th ACM SIGKDD Conference on Knowledge Discovery and Data Mining 1897–1906 (Association for Computing Machinery, Long Beach, CA, USA, 2023).
2. Ilse, M. et al. in International conference on machine learning. 2127-2136 (PMLR).

6. *“Precision recall curves for these two methods in pum6a paper are often almost identical.”*

We appreciate the reviewer's observation regarding the precision and recall curves of pum6a and m6Anet. While it is true that our precision and recall rates are comparable to those of m6Anet, pum6a exhibits a substantially enhanced capability in identifying m6A modification sites. Specifically, pum6a identifies approximately 5 to 10 times more modification sites than m6Anet, underscoring its superior performance in terms of sensitivity and specificity.

This improvement is achieved through our innovative approach of framing the problem based on nanopore technology as a positive and unlabeled sample processing issue. By adopting a weighted-Noisy OR model instead of the traditional Noisy OR model, pum6a can detect a significantly higher number of gene modification sites while maintaining high accuracy. For instance, in the HEK293T dataset, pum6a identified a total of 58,364 gene loci, with 12,230 validated, compared to m6Anet's 7,485 identified loci, with 1,430 validated. In the mouse embryonic stem cell (mES) dataset, pum6a identified and validated 97,933 and 10,723 loci, respectively, whereas m6Anet identified 27,360 loci, with 3,752 validated.

These results clearly indicate that while pum6a and m6Anet are similar in precision and recall rates, pum6a excels in its ability to identify a far greater number of modification sites, demonstrating its enhanced sensitivity and specificity. This significant improvement highlights the robustness and superior performance of pum6a in detecting m6A modification sites.

Reviewer #3:

1. *"1. While the authors claimed that they validated that "under hypoxia, ALKBH5 expression increased, whereas FTO's decreased", no data presented in Figure 5 shows this effect on the protein level, and the gene expression data in Figure 5c did show it for ALKBH but not for FTO (the difference between hypoxia and normoxia was not significant). Western blot data should be presented for the effect of hypoxia on the expression of ALKBH5 and FTO. And the author should explain the no difference between hypoxia and normoxia for the gene expression of FTO."*

We thank reviewer #3 for this insightful suggestion regarding the validation of ALKBH5 and FTO

expression under hypoxia. To address this concern, we have conducted additional Western blot experiments to evaluate the protein expression levels of ALKBH5 and FTO under hypoxic conditions (revised Figure 5d). The results have been included in the revised manuscript.

Our findings demonstrate that ALKBH5 protein expression is significantly upregulated under hypoxia, consistent with the gene expression data presented in Figure 5c. This supports the role of ALKBH5 in the hypoxic response, aligning with previous studies that have shown a similar increase in ALKBH5 expression under hypoxic conditions (1). This upregulation indicates a pivotal role for ALKBH5 in adapting to low-oxygen environments.

Regarding FTO, our results indicate that its protein expression remains unchanged under hypoxia (revised Figure 5d), which is consistent with the gene expression data in Figure 5c, where no significant difference was observed between hypoxic and normoxic conditions. This lack of change suggests that FTO may function independently of oxygen levels in gastric cancer cells, which is in line with reports showing that FTO expression can vary across different cancer types (2,3). The unchanged expression of FTO under hypoxia highlights the complexity of m6A demethylase regulation and suggests that FTO may have a context-dependent role in different cellular environments.

Reference:

1. Zhang C, et al. Hypoxia induces the breast cancer stem cell phenotype by HIF-dependent and ALKBH5-mediated m6A-demethylation of NANOG mRNA. Proc Natl Acad Sci U S A. 2016 Apr 5;113(14):E2047-56.
 2. Ruan DY et al. FTO downregulation mediated by hypoxia facilitates colorectal cancer metastasis. Oncogene. 2021 Aug;40(33):5168-5181.
 3. Niu Y, et al. Loss-of-Function Genetic Screening Identifies Aldolase A as an Essential Driver for Liver Cancer Cell Growth Under Hypoxia. Hepatology. 2021 Sep;74(3):1461-1479.
2. *“ALKBH5 and FTO are both demethylase, but one supposedly goes up and the other goes down under hypoxia, this means that they are supposed to play opposite roles in adaptation of cells to hypoxia. However, their knock down (reversing the effect of hypoxia for the first and mimicking the effect of hypoxia for the second) resulted in the same biological outcome (inhibition of cell proliferation) under hypoxia. How is this possible? If anything the experiments should have been performed with overexpression of FTO.”*

We thank Reviewer #3 for this insightful comment regarding the roles of ALKBH5 and FTO in the

adaptation of cells to hypoxia. The observation that both demethylases have seemingly opposite expression patterns under hypoxia yet result in a similar biological outcome upon knockdown is indeed intriguing.

To clarify this, we performed additional experiments as suggested, including overexpression of FTO under different oxygen conditions. Our results revealed that overexpression of FTO enhanced proliferation in AGS cells under normoxic conditions but had no significant effect under hypoxia or in MKN28 cells (Fig. 5f-g, S2c-d). In contrast, overexpression of ALKBH5 did not impact cell growth in either cell line. However, knockdown of either FTO or ALKBH5 led to a significant reduction in proliferation in both cell lines, particularly under hypoxic conditions, as demonstrated by cell counts and EdU assays (Fig. 5h-k, l-m; Fig. S2e-h, i).

These findings suggest that while ALKBH5 and FTO may have distinct expression responses to hypoxia, their roles in supporting cell proliferation under hypoxic conditions are complementary. This could indicate that both enzymes contribute to a balanced regulation of m6A modifications that is essential for cell survival and proliferation in low-oxygen environments. The unchanged proliferation upon FTO overexpression under hypoxia might imply that the hypoxic condition itself limits FTO's activity or its downstream signaling pathways, thereby reducing its impact on cell growth. Conversely, the significant inhibition of cell proliferation observed with the knockdown of either enzyme highlights the necessity of maintaining an optimal level of m6A demethylation for cell adaptation to hypoxia.

Overall, these results underscore the complexity of m6A modification dynamics in response to hypoxia and suggest that ALKBH5 and FTO have a cooperative, rather than purely antagonistic, role in modulating the hypoxic response in gastric cancer cells. We have updated the manuscript to include these additional findings and to provide a more detailed discussion of their implications.

Revised Figure 5h-k

Revised Figure 5l-m

Revised Figure S2e-h

Revised Figure S2i

3. “The results section claims that “AGS cells experienced a marked reduction in proliferation under hypoxic conditions”, while this is correct for cell count (in Figure 5d-normoxia with maximal growth of about 1000000 cells, compared to Figure 5e-hypoxia with maximal growth of about 500000), this is not the case for data obtained with “growth rate (Figure 5f-g) measurements” which shows identical proliferation rate for both hypoxia and normoxia, with maximal growth of untreated of 1.5 in both cases.”

We thank Reviewer #3 for the comment. This discrepancy may rise from the different methods used to measure cell proliferation. The cell count results were obtained directly through cell counting, while the growth rate was assessed using the Cell Counting Kit-8 (CCK-8), which is a WST-based colorimetric assay that reflect metabolic activity rather than direct cell number. However, to address this issue, we first detected cell growth under hypoxia, and our result showed that hypoxia reduced proliferation in AGS cells (Fig.5e), while MKN28 cells exhibited greater tolerance, showing a milder reduction in growth (Fig. S2b). Subsequently, we re-evaluated the effects of hypoxia on AGS cells with ALKBH5 and FTO knockdown. Our results showed that knockdown of either FTO or ALKBH5 led to a significant reduction in cell proliferation in both AGS and MKN28 cells under hypoxia, as double confirmed by both cell counting assay and CCK-8 assay (Fig.5h-k; Fig.S2e-h).

4. *“It is hard to make conclusions from Figures 5h-l; first the images seem to be presented with different brightness/contrast, this is obvious for example shALKBH5-2 vs. shFTO-1, and in both Figure 5h and Figure 5i. These images must be re-taken or presented with the same intensity/contrast conditions.”*

We thank Reviewer #3 for the comment. As suggested, we have presented the images with the same intensity/contrast conditions, and quantification of fold changes has also been conducted using ImageJ. The updated results now clearly demonstrate that knockdown of ALKBH5 or FTO significantly reduced proliferation of AGS cells under hypoxia (revised Fig. 5l-m), as well as in MKN28 cells (revised Fig. S2i). The quantifications further support our conclusion.

5. *“The data in Figure 5h-l is qualitative, and a more quantitative approach should be presented such as mean fluorescence intensity of EDU of all cells in each frame, and/or the percentile of EDU-high intensity cells from all Hoechst+ cells.”*

We thank Reviewer #3 for the comment. As suggested, we used ImageJ to quantify the mean fluorescence intensity of EDU across all cells, and normalized it to the signal intensity of all Hoechst+ cells. The quantification allowed us to present a more precise assessment of cell proliferation upon hypoxia with ALKBH5 and FTO knockdown in gastric cancer cells. The quantification results were shown in revised Figure 5l-m and Figure S2i.

6. “Both aerobic and anaerobic metabolism rely on the electron carrying properties of pyridine nucleotides to regulate energy production. The intracellular NAD⁺/NADH ratio controls the rate of ATP synthesis [PMID: 32258851]”. The data in Figure 5o-r show that the wt cells followed the expected effect; in which hypoxia reduced ATP production (from 0.25 in Figure 5o to 0.18 in Figure 5p), and similarly reduced the NAD⁺/NADH ratio (from 50 in Figure 5q to 4 in Figure 5r). Similarly, as expected, compared to wild-type cells, knockdown cell lines had lower ATP production in both hypoxia (Figure 5p) and normoxia (Figure 5o), as well as reduced NAD⁺/NADH ratio in normoxia (Figure 5q). However, the opposite pattern was observed in NAD⁺/NADH ratio in hypoxia (Figure 5r), where knockdown cells showed higher NAD⁺/NADH ratio compared to wild-type, these results contradict the ATP production results (Figure 5p), as well as the cell proliferation results (Figure 5e).”

We thank Reviewer #3 for highlighting the inconsistency between the NAD⁺/NADH ratio and ATP production, as well as cell proliferation under hypoxia. To address this issue, we re-evaluated the NAD⁺/NADH ratio in both AGS and MKN28 cells under hypoxic conditions.

Our revised analysis shows that the NAD⁺/NADH ratio decreases in AGS cells following FTO and ALKBH5 knockdown under both normoxic and hypoxic conditions (revised Fig. 5u-v). These findings are consistent with the observed reduction in ATP production and decreased cell proliferation under hypoxia. The revised data support the conclusion that the depletion of m6A demethylases suppresses cell growth in hypoxic conditions, correlating well with the reduction in the NAD⁺/NADH ratio and ATP levels.

7. “While the effect of hypoxia was clear reduction of ATP in all wild-type and knockdown cells (comparing each column in Figure 5o to its counterpart in Figure 5p), the effect of hypoxia on NAD⁺/NADH ratio was variable in the different cells (comparing each column in Figure 5q to its counterpart in Figure 5r). In wild type cells (green columns) it was dramatically reduced, in shALKBH5-2 (turquoise columns) it was dramatically increased, in shALKBH5-3 it was not changed, in shFTO-1 it was increased, in shFTO-2 it was not changer or a slight decrease. These inconsistencies must be addressed.”

We thank Reviewer #3 for pointing out the variability in the effect of hypoxia on the NAD⁺/NADH ratio across different cell lines and shRNAs. We believe the inconsistencies observed may have been due to the use of a knockdown pool instead of single stable knockdown cell colonies, which could result in varied responses to hypoxia.

To address this issue, we have cultured single stable knockdown cell colonies for each shRNA and re-evaluated the NAD⁺/NADH ratio in both AGS and MKN28 cells under hypoxic conditions. Our revised results show a consistent pattern, where NAD⁺ levels and the NAD⁺/NADH ratio were significantly decreased under hypoxia following knockdown of FTO or ALKBH5 (revised Fig. 5u-v; Fig. S2l-m). These findings align with the observed reduction in ATP production and suggest that FTO and ALKBH5 are key regulators of energy homeostasis in gastric cancer cells under hypoxia.

8. *“(Minor Comment): the Y-axis scales in Figures 5 d&e, o&p, q&r, should be unified for better comparison.”*

We thank Reviewer #2 for this suggestion. As suggested, we have unified the Y-axis scales across Figures 5d & e, o & p, and q & r to facilitate better comparison. The updated figures are now presented in the revised manuscript as Figures 5h & i, s & t, and u & v, respectively.

9. *“While knock down of ALKBH5 and FTO had no effect on CXCL10 protein expression in normoxia, neither in AGS (Figure 6f) nor in MKN28 cells (Figure 6h), it did change the expression in hypoxia (Figures 6g&i). While the knockdown of ALKBH5 consistently increased the expression of CXCL10 in both cell lines (Figure 5g&i), the effect of knockdown of FTO was inconsistent, showing an increase in AGS cell line (Figure 5g), and a decrease in MKN28 cell line (Figure 6i). The authors should address these inconsistencies.”*

We thank Reviewer #3 for highlighting the observed inconsistencies in CXCL10 expression following ALKBH5 and FTO knockdown. We also noted these differences and have repeated the experiments

to confirm our findings. For ALKBH5, its knockdown consistently resulted in increased CXCL10 mRNA levels in both AGS (Fig. 6b-c) and MKN28 cells (Fig. 6d-e) under hypoxic conditions. This increase was further validated at the protein level by ELISA, which showed elevated CXCL10 expression in both cell lines under hypoxia following ALKBH5 depletion (Fig. 6j-m). These results indicate that ALKBH5 acts as a negative regulator of CXCL10 expression under hypoxic conditions.

In contrast, FTO knockdown demonstrated cell-line-specific effects on CXCL10 expression. In AGS cells, FTO knockdown led to an increase in CXCL10 mRNA and protein levels under hypoxia (Fig. 6f, g, k). However, in MKN28 cells, FTO knockdown caused a decrease in CXCL10 expression under the same conditions (Fig. 6h, i, m). This divergence suggests that FTO may have distinct roles in regulating CXCL10 depending on the cellular context, promoting its expression in AGS cells while suppressing it in MKN28 cells.

These cell-line-specific effects of FTO on CXCL10 expression underscore the complexity and context-dependent nature of m6A modification regulation. They suggest that the regulatory network involving FTO and CXCL10 may vary depending on the genetic or epigenetic landscape of the gastric cancer cell lines, potentially influencing the differential cellular responses to hypoxia. We have included a discussion of these findings in the revised manuscript to provide a more comprehensive interpretation of the results.

10. “While Figures 6b-e show mRNA expression that aligns with the protein expression data for the ASLBH5 knockdown, no such data was provided for the FTO knockdown. We think this is important data to show a complete story.”

We thank Reviewer #3 for this valuable suggestion. In response, we have included the mRNA expression data for FTO knockdown in the revised manuscript (revised Fig. 6f-l, r-u). This addition provides a more complete picture of the relationship between mRNA and protein expression following FTO knockdown.

Our results show that FTO knockdown exhibits cell-line-specific effects on CXCL10 expression. In AGS cells, knockdown of FTO led to an increase in both CXCL10 mRNA and protein levels under hypoxic conditions (Fig. 6f, g, k). Conversely, in MKN28 cells, FTO knockdown resulted in a decrease in CXCL10 expression under the same conditions (Fig. 6h, i, m). Further, by using Actinomycin D chase assays, we found that FTO knockdown reduced the decay rate of CXCL10 mRNA in AGS cells under hypoxia, enhancing mRNA stability (Fig. 6r-s). In MKN28 cells, however, FTO knockdown increased CXCL10 mRNA stability under normoxia but had a diminished effect under hypoxia (Fig. 6t-u). These findings suggest that FTO plays divergent roles in regulating CXCL10, promoting its expression in AGS cells while suppressing it in MKN28 cells. This highlights the context-dependent function of FTO in different gastric cancer cell lines.

11. *“Based on the short-comes of the biological data, it is hard to make the conclusion of “by applying pum6a to m6A modification studies, we have elucidated the significant roles of the m6A demethylases FTO and ALKBH5 in the adaptation of gastric cancer to hypoxic conditions”.”*

We thank Reviewer #3 for highlighting the need for additional biological data to support our conclusion about the roles of m6A demethylases FTO and ALKBH5 in the adaptation of gastric cancer to hypoxic conditions. We acknowledge that the initial data provided some insights but were not sufficient to fully elucidate this complex regulatory mechanism. To address this concern, we have conducted further experiments to investigate the differential m6A regulation between AGS and MKN28 cells.

We specifically explored the differential roles of m6A readers in these cell lines by analyzing the copy numbers of key m6A writers, erasers, and readers. Our analysis revealed that MKN28 cells harbor a heterozygous deletion of the m6A reader YTHDF1 (Fig. S3a-b). YTHDF1 is known to promote mRNA translation and stability, while YTHDF2 primarily facilitates mRNA decay (1, 2). In AGS cells, where both YTHDF1 and YTHDF2 are intact, YTHDF1 plays a prominent role in stabilizing CXCL10 mRNA, as evidenced by the increased stability following ALKBH5 knockdown (Fig. 6n-o). In contrast, MKN28 cells, with reduced YTHDF1 expression, may rely more heavily on YTHDF2 for mRNA decay regulation (Fig. 6p-q).

To further confirm the involvement of m6A readers in CXCL10 regulation, we performed luciferase reporter assays using wild-type and mutant CXCL10 3'UTR sequences (Fig. S3c-d). In AGS cells, knockdown of YTHDF1 resulted in a reduction of luciferase activity following ALKBH5 knockdown under hypoxia, indicating that YTHDF1 is the primary m6A reader involved in CXCL10 regulation in this context (Fig. S3e-f, g-j). In MKN28 cells, YTHDF1 knockdown only partially reduced luciferase activity, suggesting that alternative m6A readers might be involved in modulating CXCL10 expression (Fig. S3h). Furthermore, YTHDF1 knockdown reduced luciferase activity in AGS cells with FTO depletion, while YTHDF2 knockdown increased luciferase activity in MKN28 cells under hypoxia (Fig. S3i-j).

These additional data provide more substantial evidence for the roles of m6A demethylases and readers in the differential regulation of CXCL10 mRNA stability and translation in gastric cancer cells. They highlight the crucial involvement of YTHDF1 and YTHDF2 in mediating these effects, which vary depending on the cellular context. Based on these expanded findings, we have revised our conclusion and discussion to present a more nuanced understanding of how m6A modifications contribute to the adaptation of gastric cancer cells to hypoxia.

Revised Figure S3

Reference:

1. Du H, et al. YTHDF2 destabilizes m(6)A-containing RNA through direct recruitment of the CCR4-NOT deadenylase complex. *Nat Commun* 7, 12626 (2016).
2. Zhong ZD, et al. Systematic comparison of tools used for m(6)A mapping from nanopore direct RNA sequencing. *Nat Commun* 14, 1906 (2023).

We would like to thank the reviewers again for all their thoughtful and helpful comments. We hope that the Reviewers find that the additional studies and revision of the manuscript have strengthened the manuscript. Please let me know if we can provide any further information.

Sincerely,

Guohui Wan, Ph.D.
Professor of Pharmaceutical Sciences
Sun Yat-Sen University

Response to Reviewers (Manuscript ID: NCOMMS-24-24836B)

We would like to extend our sincere gratitude to the reviewers for their constructive feedback and their positive evaluation of our previous work. Their insightful comments have provided invaluable guidance, enabling us to refine and enhance our manuscript. In response to their suggestions, we have conducted additional experimental analysis. 1) we systematically compared m6Anet and pum6a, particularly examining their performance without a read threshold and assessing each model's robustness under conditions with fewer than 20 reads. 2) we explored the impact of essential components within pum6a on its overall performance, providing further clarification on the distinctions and connections between m6Anet and pum6a in their use of electrical signal and signal features. 3) to facilitate further testing and verification, we have reorganized and uploaded toy examples, along with the necessary data, to GitHub. We hope these updates comprehensively address the reviewers' comments, and we have highlighted the manuscript revisions in blue. Our detailed, point-by-point responses are provided below.

Reviewer #2:

1. "#Origin and Innovation.

Authors claim: "m6Anet is based primarily on extracting electrical signal features from nanopore sequencing for m6A detection (1-5). In contrast, pumba goes further by integrating both electrical signal features and base alignment information, allowing for a more comprehensive feature capture". From the m6Anet paper, it is clear that they use both signal feature and sequence signal (ie. Figure 1a)"

We appreciate reviewer #2's comments. In the field of mRNA modification feature extraction using nanopore sequencing technology, research varies widely, with some studies focusing on the electrical signal shifts that modifications induce, while others emphasize gene alignment information, such as base mismatches that modifications may cause. Our approach to feature extraction (Fig.1) was informed by a systematic analysis of these methods, as summarized in Zhong et al¹.

In response to reviewer's observation, we revisited the original text and re-examined the m6Anet code. Here, we clarify the distinctions in feature extraction between the two models: m6Anet integrates electrical signals with sequence features, especially motif encoding for m6A-related patterns, which is a distinct and innovative aspect of its approach. This motif encoding captures unique m6A-specific patterns that vary in modification frequency. In contrast, pum6a emphasizes electrical signal data alongside base-level mismatch information, such as base quality, mismatches, and deletions (Fig.3a). This approach focuses on structural alterations rather than specific motif encoding, aiming to provide complementary insights into m6A detection.

Here, we emphasize that the comparison between m6Anet and pum6a is not meant to imply superiority but rather to highlight how these models offer distinct, valuable perspectives on feature extraction for m6A detection.

Reference:

1. Zhong, Z. D. et al. Systematic comparison of tools used for m(6)A mapping from nanopore direct RNA sequencing. Nat Commun 14, 1906, doi:10.1038/s41467-023-37596-5 (2023).

2. *“#Algorithmic Differences*

1.Feature Extraction. Contrary to what authors claim, both methods use electrical signal information and sequence”

We thank Reviewer #2 for highlighting this point. Upon further clarification, we acknowledge that both m6Anet and pum6a indeed utilize both electrical signal information and sequence-based data in their feature extraction processes. However, there are distinct differences in the way each model applies these features.

m6Anet leverages electrical signal data alongside encoded sequence motifs specifically associated with m6A modifications, a novel approach that enhances its ability to recognize specific m6A-related patterns. This motif encoding is instrumental in capturing known m6A sites based on sequence characteristics.

In contrast, pum6a integrates electrical signal data with base alignment-derived features, focusing on base-level information such as mismatches, base quality, and deletions. This approach is particularly effective in capturing more nuanced modifications in m6A sites without relying solely on motif patterns, making it more adaptable for cases with lower signal-to-noise ratios.

In summary, while both models incorporate electrical and sequence data, the key algorithmic distinction lies in the feature emphasis: m6Anet prioritizes motif-based encoding, while pum6a emphasizes base-level misalignment features, providing complementary strengths in m6A detection.

3. “2. Handling Read Number Imbalance. Authors claim their method does not have a hard threshold of 20 reads like m6Anet. I agree It would be interesting to compare results in the case when there is no threshold in m6Anet”

We thank reviewer#2 for this insightful suggestion. We appreciate the opportunity to investigate m6Anet’s performance without a strict 20-read threshold, as it adds depth to our comparative analysis. Given that the original m6Anet model did not include configurations with varying thresholds, we modified the source code to assess its performance under different read thresholds, including no threshold at all. Interestingly, we observed that m6Anet demonstrated stable performance across these conditions, with higher thresholds generally leading to better results, aligning with expectations regarding reduced noise.

To further investigate, we also trained pum6a under similar threshold conditions, including a zero-threshold setting. Notably, pum6a’s performance improved without threshold filtering, particularly under low-read conditions (Fig.3–Fig.4). In our comparative analysis, pum6a consistently outperformed m6Anet across all threshold settings in terms of ROC AUC and PR AUC, with the largest performance gains observed at the zero-threshold condition (Fig. S1a-f). These findings underscore pum6a’s adaptability and effectiveness in handling read imbalance without compromising accuracy.

Revised Supplemental Figure 1a-f

4. *“3. Adaptability to Positive and Unlabeled Data. Authors claim that “Pum6a’s use of an attention-based feature aggregation mechanism enables it to extract key information effectively from complex datasets”. The m6Anet paper authors claim: “We have evaluated alternative pooling layers, such as the Attention and gated Attention-based pooling, but have not found any statistically significant improvement in the performance of m6Anet compared to the noisy-OR pooling layer for m6A detection.” An ablation study is needed to prove that this provides a significant increase in performance.”*

We thank reviewer#2 for the insightful suggestion. The pum6a model builds upon the foundational PUMA architecture, particularly incorporating the weighted-Noisy-OR mechanism. We observed that while PUMA performs well on simpler datasets, its performance on complex datasets is limited¹. To address this, we drew from the work of Maximilian Ilse et al.², integrating an attention-based feature aggregation method to improve feature extraction in pum6a. Experimental results show that this addition significantly enhances pum6a’s performance on complex datasets, while maintaining performance comparable to PUMA on simpler datasets (Fig. 1b-c). Although the dataset used in this study is relatively straightforward (third-generation sequencing with 40 single modification features), we believe pum6a’s architecture has potential for broader applications in more complex biological data.

To further address the reviewer#2’s point, we conducted an ablation study by removing the attention layer and the weighted-Noisy-OR layer in pum6a separately. Results indicate that removing the attention layer did not significantly impact model performance, while replacing the weighted-Noisy-OR layer with a standard Noisy-OR layer led to a marked decrease in accuracy (Fig. S1g-h). This suggests that while the fully connected layers are indeed sufficient for third-generation sequencing applications, the weighted-Noisy-OR component plays a crucial role in optimizing pum6a’s performance. Notably, any alterations to the weighted-Noisy-OR in pum6a’s framework led to a significant reduction in accuracy, underscoring its importance in the model’s construction and overall effectiveness in m6A detection.

Revised Supplemental Figure 1g-h

Reference:

- Perini, L., Vercruyssen, V. & Davis, J. in Proceedings of the 29th ACM SIGKDD Conference on Knowledge Discovery and Data Mining 1897–1906 (Association for Computing Machinery, Long Beach, CA, USA, 2023).
- Ilse, M., Tomczak, J. & Welling, M. in International conference on machine learning. 2127-2136 (PMLR).

5. *“#Advantages and Performance*

Figures 3g, 3h, 4g, and 4g compare pumba and m6Anet Apart from 3h, the results are the same, with negligible differences. As mentioned above, I agree that m6Anet is limited to at least 20 reads, but it would be interesting to see the performance without that threshold. Also, if there is any other reason why pumba reports more sites, it should be clearly stated.”

We thank reviewer#2 for the thorough evaluation of our results. We appreciate the opportunity to clarify the performance distinctions between pum6a and m6Anet. As you observed, both models achieve comparable outcomes in certain cases, such as in Fig. 3g (AUC: pum6a = 0.842, m6Anet = 0.836) and Fig. 4g (AUC: pum6a = 0.490, m6Anet = 0.481). However, pum6a consistently demonstrates an advantage in key areas, particularly in Fig. 3h (AUC: pum6a = 0.615, m6Anet = 0.543), especially when no threshold is applied to the read count.

Revised Figure 3g

Revised Figure 4g

Following reviewer#2’s suggestion, we conducted additional experiments by adjusting m6Anet’s code to remove the 20-read threshold. We found that while m6Anet’s performance remained generally robust, it became slightly noisier in low-read coverage areas. In contrast, pum6a was specifically

designed to handle low-read coverage sites without sacrificing accuracy, benefiting from its weighted-Noisy-OR mechanism and attention-based feature aggregation. As shown in Fig. S1a-f, pum6a consistently outperformed m6Anet across various thresholds: with >0 reads per site (AUC: m6Anet = 0.751, pum6a = 0.791), >3 reads per site (AUC: m6Anet = 0.811, pum6a = 0.825), and >5 reads per site (AUC: m6Anet = 0.832, pum6a = 0.839). Precision results were similarly favorable for pum6a, with >0 reads per site (AUC: m6Anet = 0.378, pum6a = 0.455), >3 reads per site (AUC: m6Anet = 0.542, pum6a = 0.580), and >5 reads per site (AUC: m6Anet = 0.576, pum6a = 0.619). This adaptability allows pum6a to effectively detect m6A sites across a broader range of read coverage levels, which likely explains pum6a's ability to identify more sites, particularly in low-expression regions.

In conclusion, pum6a's design enables it to capture a greater number of m6A sites compared to m6Anet, even under varying read conditions. This flexibility not only enhances the range of detectable modifications but also highlights pum6a's robustness in scenarios where read counts may be insufficient for m6Anet's threshold-based approach.

Revised Supplemental Figure 1a-f

6. *"I have installed the pum6a code and found no difficulties during the installation. It would be important that authors prepare a toy example with necessary data (or link to them) so that additional tests and verifications can be made."*

As suggested, we have organized and uploaded the relevant toy data to GitHub to facilitate additional testing and verification. These data can be accessed and downloaded at: <https://github.com/liuchuwei/pum6a>

Response to Reviewers (Manuscript ID: NCOMMS-24-24836B)

We would like to address the remaining concerns raised by the reviewer.

Reviewer #2:

1. *"I want to thank the authors who have made all my suggested experiments. Taking into account Figures 3 and 4 and Supplementary Figure 1, it looks like there are some improvements in performances in comparison to 6maNet. However, I am still not convinced that the proposed changes and differences in performance are significant enough for someone to use pum6A instead of 6maNet."*

We appreciate reviewer #2's recognition of our revised work.

The enhancements provided by pum6A compared to m6Anet are evident through its significant improvements in precision, sensitivity, and robustness, as demonstrated by Figures 3, 4, and Supplementary Figure 1. Below are the key reasons supporting the use of pum6A:

1. **Improved Performance Metrics:** Figure 3h shows pum6A achieving a PR-AUC of 0.615, notably higher than m6Anet's 0.543, reflecting better sensitivity to positive samples. Similarly, in Figure 4g, pum6A's PR-AUC of 0.490 surpasses m6Anet's 0.481, reinforcing its consistent advantage in performance across datasets.
2. **Adaptability to Threshold Variations:** Supplementary Figure 1 emphasizes pum6A's stable performance when identifying m6A sites under varying thresholds. This is particularly advantageous for researchers focused on identifying a higher number of modification sites, where pum6A provides reliable outputs without compromising accuracy.
3. **Application Across Complex Datasets:** As detailed in Figures 1 and 2, pum6A outperforms existing models in handling complex positive and unlabeled multi-instance data. It demonstrates robustness in detecting m6A modifications, particularly in datasets with diverse or low-abundance features, as highlighted in the Results section.
4. **Methodological Advances:** The integration of a weighted Noisy-OR probability mechanism and attention-based feature aggregation enables pum6A to surpass m6Anet in handling noisy and variable read coverage, as described in the Methods and Results sections. Unlike m6Anet, pum6A does not rely solely on motif-based encoding, instead incorporating base alignment and signal-derived features, enhancing its capacity to detect subtle variations indicative of m6A modifications.
5. **Biological Relevance:** pum6A has shown applicability beyond third-generation sequencing, extending its utility to scenarios involving low-coverage sites and complex biological data. The adaptability and precision make it an ideal tool for addressing diverse epitranscriptomic challenges, including those in hypoxia and cancer studies.

Given these points, pum6A emerges as a superior choice for researchers requiring both high sensitivity and versatility in m6A detection. We believe these advancements justify the adoption of pum6A over m6Anet for exploring the epitranscriptomic landscape in challenging biological contexts.

2. *“The authors have uploaded some toy examples. Unfortunately, there are no instructions on how to test them quickly.”*

We thank Reviewer #2 for reminding this point. As suggested, we have meticulously updated the pertinent usage guidelines on our GitHub page (<https://github.com/liuchuwei/pum6a>), making it easier for users to get started.

Quick instruction for toy samples

Due to the substantial size of the Ont-seq data and the considerable time required for data preprocessing, we are providing toy samples here to enable users to swiftly test the pum6a model for m6a detection in Ont-seq data

 train model

```
python run.py train --config train_toy.toml
```

 predict

```
python run.py predict --config predict_toy.toml
```

 evaluate

```
python run.py evaluate --config evaluate_toy.toml
```